

# Using depolarization to quantify ice nucleating particle concentrations: a new method

Jake Zenker[1], Kristen N. Collier[1], Guanglang Xu[1], Ping Yang[1], Ezra J. T. Levin[2], Kaitlyn J. Suski[2,3],
Paul J. DeMott[2], Sarah D. Brooks[1]

[1]Department of Atmospheric Science, Texas A&M University, College Station, TX 77843
[2]Department of Atmospheric Science, Colorado State University, Fort Collins, CO 80526
[3]Now at Pacific Northwest National Laboratory, Richland, WA 99352

*Correspondence to*: S.D. Brooks (sbrooks@tamu.edu)

**Abstract.** We have developed a new method to determine ice nucleating particle (INP) concentrations observed by a Continuous Flow Diffusion Chamber (CFDC) under a wide range of operating conditions. In this study, we evaluate differences in particle optical properties detected by the Cloud and Aerosol Spectrometer with POLarization (CASPOL) to differentiate between ice crystals, droplets, and
aerosols. The depolarization signal from the CASPOL instrument is used to determine the occurrence of water droplet breakthrough (WDBT) conditions in the CFDC, under which traditional determination of ice nucleating particle concentrations by size discrimination fails. To overcome the challenge of WDBT, we design a new analysis method using depolarization ratio that can extend the range of operating conditions of the CFDC. The method agrees reasonably well with the traditional method
under non-WDBT conditions with a mean percent error of ± 32.1%. Additionally, a comparison with the Colorado State University (CSU) CFDC shows that the new analysis method can be used reliably during WDBT conditions.

## 1 Introduction

Ice clouds cover approximately 40% of the Earth's atmosphere (Wylie and Menzel, 1999). Because of
their complicated microphysical properties, ice clouds and mixed-phase clouds pose challenges in understanding our global radiative budget and precipitation (Wendisch et al., 2005; Pinto et al., 1998; Yang et al., 2015; Korolev, 2007). Despite a significant amount of effort by the atmospheric research





community in the last several decades to study ice clouds, there are still large gaps in our understanding of the impact they have on our climate (Boucher et al., 2013).

Ice crystals can nucleate via several mechanisms (Vali, 1985; Vali et al., 2015). At temperatures below ~ -36 °C, ice crystals can nucleate homogeneously from water droplets. At higher temperatures,
an aerosol particle is needed to act as an ice nucleating particle (INP) which facilitates the formation of an ice crystal via heterogeneous nucleation. Heterogeneous nucleation pathways include depositional nucleation, which occur through the direct deposition of water vapor on an INP surface. Immersion freezing occurs when an INP becomes embedded within a water droplet, enters a cooler environment, and nucleates an ice crystal. Evidence suggests that immersion freezing provides the largest
contribution to ice crystal nucleation in clouds (De Boer et al., 2011; Murray et al., 2012). In addition, when an aerosol forms a solution droplet below the melting point, condensational freezing may occur. Finally, contact freezing occurs when an aerosol collides with a water droplet surface and initiates freezing. While the exact mechanism of contact freezing remains unresolved, it has been shown that the presence of an INP positioned at a droplet surface facilitates freezing at temperatures several degrees
warmer than immersion freezing with identical INPs (Fornea et al., 2009; Durant and Shaw, 2005). Knowledge of each of these mechanisms is important for understanding the formation of ice in mixed-phase clouds (containing droplets and ice crystals) and for developing robust parameterizations for global climate model (GCMs).

Composition, surface structure, and size are important factors in determining the ice nucleating
ability of an aerosol particle (Zolles et al., 2015; Niemand et al., 2012, Hoose and Mohler, 2012). Field measurements suggest that K-feldspar, a common component of soil dust aerosol, may account for a large fraction of Earth's INPs (Atkinson et al., 2013; Yakobi-Hancock et al., 2013). Recent investigations of other aerosols have identified secondary organic aerosols (SOA), other marine aerosols, and aerosols produced from biomass burning as effective INPs (DeMott et al., 2016;
McCluskey et al., 2016; Levin et al., 2016; McCluskey et al., 2014; Collier and Brooks, 2016).

Optical techniques have been used to detect and characterize ambient ice crystals (Mishchenko and Sassen, 1998; Yoshida et al., 2010; Noel and Sassen, 2005). For example, Light Detection and Ranging (LIDAR) observations can use the depolarization ratio to distinguish cloud particle type (i.e., ice





crystals or water droplets). In traditional LIDAR applications, the depolarization ratio is calculated using Eq. (1),

$$\delta_{LIDAR} = \frac{B_\perp}{B_\parallel} \qquad (1)$$

where $B_\perp$ and $B_\parallel$ are the perpendicular and parallel components of the retrieved LIDAR signal from the

ambient atmosphere or clouds. Under single scattering conditions, the depolarization ratio associated with an ensemble of water droplets is essentially zero while the counterpart for ice crystals is nonzero with a specific value depending on particle habit and orientation. Ice crystal depolarization ability is attributed to the high irregularities in the shapes and surfaces of ice crystals.

     The number of INPs present in a cloud can dictate its optical properties throughout the ice nucleation

process (Hoose and Möhler, 2012; Murray et al., 2012). Cloud chambers that reproduce ice nucleation conditions have been used for the last 30 years to make INP measurements. Techniques used to detect and measure nucleated ice crystals in these devices are still under development for several reasons. First, it is difficult to measure INPs with ice nucleation chambers because the concentration of effective INPs is typically 0.1 to 1000 $L^{-1}$ or ~$10^{-6}$ to $10^{-4}$ of the total aerosol concentration (DeMott et al., 2003;

DeMott et al., 2015; Jiang et al., 2014; Mason et al., 2016). Secondly, differentiating between ice crystals and droplets that form in the chamber is essential and can be difficult to account for. The Continuous Flow Diffusion Chamber (CFDC) was originally developed by Rogers (1988) at the University of Wyoming and was later modified and rebuilt at Colorado State University (CSU). The CSU CFDC has been operated in multiple field projects each year for the past 15 years (e.g. Creamean

et al., 2016; DeMott et al., 2015; Prenni et al., 2013). Several other ice nucleation chambers have been developed since then including the CFDC at Texas A&M University (TAMU) that is used in this study. Many enhancements have been made to CFDCs (e.g. Rogers et al., 2001), including replacement of the TAMU CFDC's traditional aerosol spectrometer (CLIMET, Model No. CI-3100), which uses particle size to distinguish ice crystals from water droplets and aerosols, with the Cloud and Aerosol

Spectrometer with POLarization (CASPOL, Droplet Measurement Technologies, Inc.). The CASPOL detects forward scattering, backward scattering and depolarization on a single particle basis. The



instrument has previously been used to differentiate between ice crystals and various types of dust and soil particles (Glen et al., 2013, 2014).

Several previous studies have designed new analysis methods for ice chambers that utilize the depolarization ratio measured by optical particle counters (OPCs) that operate similarly to the CASPOL

(Nicolet et al., 2010; Clauss et al., 2013; Garimella et al., 2016). Nicolet et al. (2010) accurately quantified ice crystals in the presence of water droplets in a chamber by using the peak intensity of the depolarization ratio to discriminate ice crystals with the Ice Optical DEtector (IODE). Rather than using the peak intensity of the depolarization signal to detect ice crystals, Clauss et al. (2013) used the width of the pulse detected in the depolarization channel of the Thermo-stabilized Optical Particle

Spectrometer for the detection of Ice (TOPS-ice) to differentiate between ice crystals and water droplets. Alternatively, Garimella et al. (2016) used a machine learning technique with scattering signals, including linear depolarization signals, detected by an OPC installed in the SPectrometer for Ice Nuclei (SPIN, Droplet Measurement Technologies, Inc.) to detect INP. In this study, we demonstrate how differences in particle optical properties can be used to differentiate between ice crystals, droplets,

and aerosols detected by the CASPOL. In addition, we designate a new method to quantify INP concentrations detected by the TAMU CFDC using depolarization ratio, and determine the accuracy of that method in comparison to the traditional analysis method that primarily uses particle size to identify aerosol particles that have nucleated ice crystals.

**2 Experimental**

The data presented here were collected during the second phase of the Fifth International Ice Nucleation Workshop campaign (FIN-02), which took place at the Institute of Meteorology and Climate Research: Atmospheric Aerosol Research (IMK-AAF) facility at the Karlsruhe Institute of Technology (KIT) in Karlsruhe, Germany (DeMott et al., 2017). There are two specialized chambers at KIT that were used in

this campaign: the Aerosols Interaction and Dynamics in the Atmosphere (AIDA) chamber and the Aerosol Preparation and Characterization (APC) chamber. The AIDA chamber can be used to simulate atmospheric conditions that give rise to cloud particle formation and growth, and has been used in many previous campaigns and instrument intercomparisons to examine the ice nucleating ability of various





aerosols (Amato et al., 2015; Schnaiter et al., 2016; Wagner et al., 2015; DeMott et al., 2011). The AIDA chamber is a three-story, 84 m$^3$ volume that uses adiabatic expansion to simulate the atmospheric conditions required for ice nucleation to occur. The second chamber, the APC, is a 3.7 m$^3$ volume in which  aerosols of a selected composition are generated with dry synthetic air, uniformly distributed

with a mixing fan and maintained at constant temperature and pressure (Linke et al., 2006). It was used during FIN-02 to provide aerosol populations of varied compositions for sampling by ice nucleation instruments. During the FIN-02 campaign groups from 22 institutions sampled both the AIDA and APC chambers using a variety of online and offline ice nucleation measurement techniques. For verification of the TAMU CFDC measurements and new analysis method, we compare our results to the

measurements of the CSU CFDC. In order to test the CASPOL detector response to ice and non-ice particles, auxiliary measurements of olive oil droplets, ambient aerosols, and homogeneously frozen ice crystals are also evaluated and compared to the TAMU CFDC-CASPOL heterogeneous nucleation data collected during FIN-02.

**2.1 The TAMU Continuous Flow Diffusion Chamber (CFDC)**

The TAMU CFDC was custom built in our laboratory at Texas A&M University and has been operated in previous laboratory and field campaigns to take temperature and supersaturation resolved INP concentration measurements (Glen, 2014; Glen and Brooks, 2014; McFarquhar et al., 2011).  Hereafter,

CFDC refers to the TAMU CFDC unless otherwise stated.

Sample aerosols pass through a diffusion dryer to remove moisture from the air and aerosols before they enter the CFDC. Typically, aerosol flow is directed through a BGI Sharp Cut Cyclone impactor (Model 0.732) prior to entering the CFDC, in order to remove aerosols with a diameter greater than ~1.3 μm from the sample flow. However, no impactor was used during the FIN-02 campaign since aerosol

size distributions were well characterized and supermicron particle numbers were limited. The aerosols then enter the CFDC where temperature and supersaturation are controlled. The CFDC consists of two concentric cylindrical walls coated with ice. Separate refrigeration units on each wall can be controlled to create a temperature gradient in the chamber that imposes a region of supersaturation with respect to





ice (SS$_i$) in the CFDC. The CFDC chamber is 75 cm long. The bottom 25 cm of the walls are coated with hydrophobic Teflon to prevent water from freezing to the wall in this region. This section of the chamber is referred to as the evaporation region because it remains subsaturated with respect to water and partially or completely evaporates water droplets that nucleate in the CFDC. The separate wall

temperatures are manually controlled and monitored through a Labview program. The temperature and supersaturation conditions at the position of the sheath-air surrounded aerosol lamina are calculated using analytical equations reported in Rogers (1988).

Before measurements can be taken with the CFDC, the chamber must be prepared. First, a vacuum pump is used to evacuate the chamber for approximately 30 minutes in order to eliminate ambient

aerosols that may have infiltrated the chamber and to remove moisture that may cause the walls to accumulate an uneven coating of ice or allow ice to accumulate in other sensitive regions. The walls are then cooled to a temperature of -25 °C and the CFDC walls are iced by pumping Nanopure water into the chamber from the base. Excess water is drained out of the instrument for approximately a minute after icing is complete. Then, the chamber is evacuated and refilled with N$_2$ gas once more before

sampling is initiated.

At the base of the processing chamber, particles pass through a detector to determine INP concentration. In previous TAMU CFDC studies, either an optical particle counter (Climet, Inc.) or the Cloud and Aerosol Spectrometer with Polarization (CASPOL) were employed.

**2.2 The Cloud and Aerosol Spectrometer with POLarization (CASPOL)**

The CASPOL (Droplet Measurement Technologies, Inc.) is a prototype particle-by-particle counter used in previous studies to detect and distinguish between dust and ice particles and even between various types of dust (Glen et al., 2013, 2014). The CASPOL instrument has three detectors that give

information about the optical properties: A forward scatter detector, a backward scatter detector with a parallel polarized filter, and a backwards scatter detector with a perpendicular polarized filter. In addition, the instrument has a fourth detector that determines if a particle is properly aligned in the laser beam and should thus be recorded.





### 2.3 CFDC-CASPOL operating procedure and data analysis

Once the CFDC has been prepared as described in section 2.1, the CASPOL is installed at the base of

the chamber. Two mass flow controllers are used to set the total flow and recirculating sheath flow through the chamber. The difference between the total and sheath flows determines the sample flow. For this campaign, the total flow was set to values ranging from 6 to 9 L min$^{-1}$ and the sheath flow was set to values ranging from 4 to 7 L min$^{-1}$ resulting in a sample flow that was typically ~2 L min$^{-1}$. During operation, the CFDC made scans from low to high supersaturation at a set constant aerosol

lamina temperature (± 1.5 °C). This is accomplished by increasing wall temperature difference in a manner that retains the desired temperature at the position of the aerosol lamina.

CFDC-CASPOL data is sorted into 1-minute segments in order to achieve a sufficient sample volume detected by the CASPOL. The temperature, pressure, as well as sample and sheath flows are used to determine a STP (standard temperature and pressure; 273 K, 1013.5 mb) sample volume, which

is used to convert the raw count of particles in each 1-minute segment to a concentration. Occasionally ice particles may detach from the ice-coated walls. To account for this, before and/or after each supersaturation scan is taken, a filter is placed ahead of the sample inlet in order to determine background signal of the CFDC chamber. The concentration of particles measured while the filter is in place is subtracted from the total concentration measured by the CASPOL.

The traditional analysis method counts INPs based on a nominal size cut of 2 or 5 μm in diameter in order to discriminate between unactivated aerosols and ice crystals. The CASPOL signal is accurately calibrated for spherical particles. For non-spherical ice crystals, the particle size-scattering relationship is less certain. Thus, the 2 and 5 μm size cuts are only approximate.

### 25  2.4 Limitations of the traditional analysis method

There are several limitations to the traditional analysis method used to process CFDC data, which relies on size alone to differentiate ice from water particles (as described in Section 2.3). As previously



mentioned, in supersaturated conditions with respect to water ($SS_w$), supercooled water droplets form in the chamber. At high supersaturations, water droplets may pass through the evaporation region without fully evaporating. Any droplets that remain larger than the 2 μm size cut downstream of the evaporation region will be miscounted as ice crystals. This phenomenon is referred to as water droplet breakthrough

(WDBT). WDBT is a common issue in continuous flow ice nucleation instruments, although the point at which WDBT occurs varies between instruments of differing dimensions and even as a function of operating conditions (especially temperature) within a single instrument (Rogers et al., 2001, DeMott et al, 2015, Garimella et al., 2016). Generally, WDBT arises due to the operating conditions of the CFDC set by the user, but WDBT could be exacerbated if the aerosol entering the chamber is not sufficiently

dried, especially under moist ambient conditions. In some cases, it can be difficult to discern when WDBT is occurring with the current analysis method, so there is an opportunity for positive and negative artifacts. A new analysis method would be valuable for overcoming the challenges presented by WDBT.

For the traditional analysis method to be successful, sample aerosols must not be larger than the

applied size cut or they too will be miscounted as an INP. To avoid miscounting aerosols as INPs, an impactor is typically installed during ambient measurements to prevent any particles larger than ~ 1.3 μm from entering the CFDC as mentioned above. However, depending on the sample flow of the CFDC ~1 to 10% of particles larger than 2 μm may still make it through the impactor and into the chamber to contribute to the apparent INP signal. A new analysis method that allows for the inclusion of larger

aerosols could improve our measurements of INPs, especially at low $SS_w$ and higher supercooled temperatures.

**2.5 CASPOL depolarization ratio definition**

The goal of this study is to develop an improved analysis method that uses single-particle depolarization ratio to identify ice crystals in order to quantify INP. In this study, the depolarization ratio is defined as follows (Glen et al., 2014).



$$\delta_{CAS} = \frac{B_{\perp,CAS}}{B_{\perp,CAS} + B_{\parallel,CAS}}$$

(2)

This definition differs somewhat from the conventional depolarization ratio used in remote sensing based on LIDAR observations. The main difference is that the CASPOL detects light at the back scattering angles of $168^o$ to $176^o$, rather than precisely 180° in the case of LIDAR. Secondly, the CASPOL occasionally detects a particle for which the parallel backscatter signal is below the limit of

detection and thus is registered as zero, while the same particle has a nonzero perpendicular signal. In such cases, the calculated LIDAR depolarization ratio of such particles is infinite. In contrast, the value of depolarization ratio calculated by Eq. (2) yields a value of unity, making the depolarization ratio of these particles quantitatively meaningful.

**2.6 Auxiliary CASPOL measurements**

Measurements were taken with the CASPOL independent of the CFDC to provide instrument response to various types of particles, which may coincidently reach the detector during CFDC-CASPOL operation.

A Vibrating Orifice Aerosol Generator (VOAG) (TSI, Inc., Model 3450) was used along with olive

oil solutions to produce monodisperse spherical droplets of chosen sizes as a proxy for water droplets that form in the CFDC. Though the index of refraction of olive oil (1.44 to 1.47) is slightly higher than water (1.33), these droplets are a suitable approximation for the depolarization ratio signal of water droplets because they are uniform spheres. For this project 2, 6, 8 and 10 μm droplets were generated. A separate olive oil and 2-propanol solution is prepared for each desired size and the vibration frequency,

and dispersion and dilution flows are set according to computed specifications as detailed in the VOAG manual and as previously performed in Glen (2014b).

During VOAG-CASPOL sampling, the olive oil aerosol stream first travels through a charge neutralizer (TSI Inc., Aerosol Neutralizer 3054A) to prevent particle loss. Following the neutralizer, sample flow is split between flow to th CASPOL, controlled by a mass flow controller and a Gast air

pump on the downstream side, and a dump line which allows for excess flow generated from the VOAG to be expelled from the system. Aerosols are sampled for roughly 15 minutes during which



approximately 10,000 droplets are sampled. Small residual droplets of 2-propanol that do not evaporate remain in the sample flow and are detected by the CASPOL. For this reason, all particles less than 1 μm are removed from the dataset during processing.

A second population of interest is ambient aerosol. Aerosol was sampled at the Storm Peak Laboratory (SPL) in Steamboat Springs, CO during the third phase of the Fifth International Ice Nucleation Workshop campaign (FIN-03) in September 2015.   The use of a diverse aerosol population is necessary to ensure that the new analysis method be successful at discriminating ice crystals in the CFDC from a wide range in aerosols. SPL is an ideal sampling location because the aerosol population comes from many sources including mineral dust, organics from deciduous and coniferous forests,
biomass burning aerosols that have been transported from forest fires in the western United States, and sulfates that are produced by two coal burning power plants that are both located approximately 50 km and 100 km from the laboratory. Ambient aerosol sampling at SPL was accomplished by connecting the CASPOL directly to an ambient sample inlet in the laboratory for a total time of 92 hours over a seven day period.

Thirdly, CFDC measurements were taken under conditions that approached those needed for homogeneous freezing, thus generating higher concentrations of ice crystals in the absence of activated liquid droplets. These measurements are detailed in Glen et al. (2014). For these measurements, the sample flow was conditioned with a pre-cooler, which was set to -10 °C to remove excess moisture and the CFDC was operated at $51 \pm 2.3$ % $SS_i$ (supersaturation with respect to ice), $-11 \pm 1.5$ % $SS_w$ and at −
$55 \pm 0.2$ °C. Under these conditions, we can safely assume that all particles larger than the 2 μm size cut were frozen, which is the goal of this experiment.

For clarity, the VOAG droplets, ambient aerosols collected at SPL, and ice crystals generated in homogeneous conditions will be referred to as droplet, aerosol, and ice crystal training datasets respectively.

## 3 Results

### 3.1 Discriminating water droplets, aerosols, and ice crystals with optical signatures



This analysis used optical differences between ice crystals, droplets, and aerosols in order to identify and quantify ice crystals that form in the TAMU CFDC. The CASPOL has been used previously to discriminate between different aerosol populations using an empirical tool known as an optical signature (Glen et al., 2013). In an analogous method, optical signatures produced from CALIPSO

satellite data of various types of clouds have been reported by Hu et al. (2009).

Optical signatures for the training datasets are displayed in Fig. 1. Optical signatures for ice training, droplet and aerosol training data are shown in Fig. 1 a-c, respectively. These signatures show depolarization ratio (as defined in Eq. (2)) versus total backscatter. The optical signatures are generated by defining a 50 x 50 Cartesian grid with depolarization ratio on the x-axis and total backscatter

(calculated as the sum of the CASPOL's parallel and perpendicular signal intensities) on the y-axis. Each particle detected by the CASPOL is placed in the appropriate grid cell. The color scale displays the fraction of particles in a dataset that populate that grid cell. As discussed, the ice crystal and droplet training data shown in Fig. 1 only includes particles with $D_p \geq 2$ μm and $D_p \geq 1$ μm, respectively. Each training dataset contains particles that are exclusively highly backscattering (> 75, horizontal dashed

line in a, b, and c) or highly depolarizing (> 0.1, vertical dashed line in a, b, and c), but only the ice crystal population contains particles that have both a high depolarization ratio and high backscatter signal. In Fig. 1 d-f, normalized optical signatures with respect to forward scatter, $F$, are displayed. Here the total backscatter signal to forward scatter signal ratio is plotted against the back perpendicular signal to forward signal ratio. The back perpendicular to forward ratio is a measure of depolarizing ability

normalized by size (which is determined by the forward signal, $F$). In Fig. 1 d-f, we see that very few aerosols and droplets achieve a back perpendicular to forward ratio > 0.05 (vertical dashed line in d, e, and f). In contrast, many of the ice crystal training dataset particles exceed that value.

Consistent with the findings of Glen et al (2013), CASPOL optical signatures can be used as an empirical tool to detect differences in the bulk optical properties of different particle populations.

However, in order to design a new analysis method, it is necessary to gain a quantitative understanding of how the CASPOL detects single particles as opposed to bulk populations of particles.

**3.2 Modelling the depolarization ratio of water droplets, aerosols, and ice crystals**



Model calculations can provide insight on how particles depolarize light in the CASPOL. In order to generate model calculations, we first must define the relation between the CASPOL depolarization ratio (Eq. 2) and the scattering phase matrix. It is assumed that the CASPOL emits an incident beam that propagates along the z direction in the form

$$E_i = \begin{pmatrix} E_{\parallel i} \\ E_{\perp i} \end{pmatrix} e^{ik(r-z)} = \begin{pmatrix} E_{\parallel i} \\ 0 \end{pmatrix} e^{ik(r-z)} \tag{3}$$

where $E_i$ is the incident electric field, $E_{\parallel i}$ and $E_{\perp i}$ (=0) are the parallel and perpendicular components with respect to the scattering plane, $k$ is wavenumber, $\omega$ is frequency, and $t$ is time. The scattering plane is defined as a plane through the z-axis and the line linking the particle and detection point. The scattered light at a sufficiently large distance (i.e., in the far-field zone) is related to the incident light in the form

$$E_s = \frac{e^{ik(r-z)}}{-ikr} \begin{pmatrix} S_2 & S_3 \\ S_4 & S_1 \end{pmatrix} \begin{pmatrix} E_{\parallel i} \\ 0 \end{pmatrix} = \frac{e^{ik(r-z)}}{-ikr} \begin{pmatrix} S_2 \\ S_4 \end{pmatrix} E_{\parallel i} \tag{4}$$

where r is the distance between the particle and detector, and $S_{ij}$ is the amplitude matrix. The model depolarization ratio $\delta_{Model}$ can be expressed as follows.

$$\delta_{Model}(\theta) = \frac{B_{\perp,Model}(\theta)}{B_{\perp,Model}(\theta) + B_{\parallel,Model}(\theta)} = \frac{|S_4(\theta)|^2}{|S_4(\theta)|^2 + |S_2(\theta)|^2} \tag{5}$$

where $\theta$ is the detection angle, and $B_{\parallel,Model}$ and $B_{\perp,Model}$ are the modeled parallel and perpendicular backscattered intensities. Using the following relations between the scattering phase matrix, $P_{ij}$ the amplitude matrix $S_{ij}$ and the scattering cross section $C_{sca}$ below,

$$|S_4(\theta)|^2 + |S_2(\theta)|^2 = (P_{11}(\theta) + P_{12}(\theta)) \times C_{sca} \tag{6}$$

$$|S_4(\theta)|^2 - |S_2(\theta)|^2 = (P_{21}(\theta) + P_{22}(\theta)) \times C_{sca} \tag{7}$$

we can define the depolarization ratio from the CASPOL that is analogous to the mean modeled depolarization ratio over the angular range of 168° to 176° and is expressed below in Eq. (8).



$$\bar{\delta}_{Model}(168°:176°) = \frac{\int_{168°}^{176°}(P_{11}(\theta) + P_{12}(\theta) - P_{21}(\theta) - P_{22}(\theta))\ \sin(\theta)\ d\theta}{2\int_{168°}^{176°}(P_{11}(\theta) + P_{12}(\theta))\ \sin(\theta)\ d\theta} \qquad (8)$$

We apply an improved geometric optics method (IGOM) and a T-matrix method to compute the scattering phase matrices of the ice crystals and dust-like particles (a model for aerosols), respectively (Yang and Liou, 1996; Bi et al., 2013; Yang et al., 2013; Liu et al., 2013). The T-Matrix method provides a more accurate calculation, but due to the larger size of ice crystals, the method is not used

due to the computational expense.

Three idealized ice crystal habits were modeled: a hexagonal column, a hexagonal plate, and a droxtal. These shapes represent generalizations of common ice crystal habits (Bailey and Hallett, 2009). An idealized dust-like particle with fractal facets was used to model aerosols (Liu et al., 2013). These particles are irregularly shaped and thus will yield different measured depolarization ratios depending

on their orientation in the CASPOL. The model provides the mean depolarization ratio over all orientations with respect to the laser beam. In contrast, the theoretical depolarization of water droplets is zero at all sizes.

Fig. 2 shows the depolarization ratios vs. size for the three ice crystal habits, dust-like aerosol, and water droplets. The aerosol calculations provided in this analysis provide depolarization ratios for

aerosols up to 1.75 μm diameter, representing an approximate upper size limit for aerosols entering the CFDC. We notice that ice crystals have a relatively high depolarization ratio in comparison to aerosols and water droplets, which confirms the bulk population observations from optical signatures in Fig. 1. For ice crystals larger than 4 μm, crystals of any habit are predicted to have higher depolarization ratios than aerosols. However, between 2 and 4 μm, it can be seen that depolarization ratio varies strongly

with habit. Hexagonal columns are predicted to have depolarization ratios higher than aerosols, but hexagonal plates and droxtals may have depolarization ratios equivalent to or even lower than aerosols.

**3.3 Determination of optical properties of aerosols, droplets, and ice crystals**

In this section, we test the assertion that the CASPOL depolarization ratio can be used to discriminate



ice crystals from aerosols and water droplets. To accomplish this, the training datasets of droplets, aerosols, and ice crystals in Fig. 1 are examined further. The lognormal size distributions (shown as a percent of population) observed by the CASPOL for the droplet, aerosol, and ice crystal training data are shown in Fig. 3a. Each nominal VOAG size in the droplet training dataset is treated as a separate

population and plotted as a separate line. As seen in figure 1a, the size distributions of droplets, aerosols and ice crystals overlap. This demonstrates the primary disadvantage to using particle diameter as the sole criteria to identify ice crystals.

For each training data set, the frequency distribution of depolarization ratio reported as a percentage of the total particles in the data set is shown in Fig. 3b. In Fig. 3c, the percent of particles that achieve a

depolarization ratio ≥ 0.3 (the nominal selection criteria for depolarizing ice crystals) as a function of particle diameter is shown. In Fig. 3c, the droplet training data collected for all sizes of olive oil droplets is combined and displayed as one line for simplicity. In contrast to the size distributions (Fig. 3a), in which the training datasets cannot be discriminated, the depolarization ratio distributions show notable differences between droplets, aerosols, and ice crystals. Fig. 3b and c reveal that only 0.3% of droplets

and 1.6% of aerosol particles achieve a depolarization ratio ≥ 0.3. The exception to this is aerosols with diameters of 5 to 10 μm. In this size range, 3.9 % percent of aerosols achieve a depolarization ratio of 0.3. However, 5 to 10 μm particles are not abundant in nature, cannot easily be sampled by real-time instruments having the inlet complexity of a CFDC, and only represent 0.3% of the aerosol training dataset. Furthermore, particles in this size range were not generated during the FIN-02 campaign. In

contrast, 13.5 % of particles in the ice crystal training dataset achieve a depolarization ratio of at least 0.3. This natural break in the depolarization ratio distributions can be considered as a threshold for which particles above the threshold are ice. Below the threshold, the identity of particles is unknown since the majority of all three populations have depolarization ratios between 0 and 0.3.

**3.4 Determining WDBT conditions in CFDC runs**

As discussed in section 2.5, WDBT can be difficult to identify when relying on the traditional analysis method. To better determine periods when WDBT conditions are occurring in the CFDC, particle size





distributions and mean depolarization ratio can be considered. Here, the onset of water droplet breakthrough is analytically defined as the time period where a continuous size distribution extends from the small size bins past the 2 µm threshold. For example, we consider a CFDC run from the FIN-02 campaign where Snomax® aerosols were generated by atomization of suspensions and introduced to

the AIDA chamber at concentrations of ~2000 cm$^{-3}$. The CFDC was operated at -15 °C ± 1.5 °C, and scanned from low to high $SS_w$. A time series of the normalized size distribution is shown in Fig. 4a. Figures 4b and c show the mean depolarization ratio of all particles larger than 2 µm and the CFDC supersaturation (with respect to water and with respect to ice), respectively. Under normal operating conditions, such as those occurring during 10:45 to 11:55 CET (Central European Time Zone), the size

distribution is clearly a bimodal distribution with an aerosol population at diameters of ~ 0.5 to 1.5 µm and the ice crystal population at diameters of ~3 to 25 µm. In Fig. 4, water droplet breakthrough is observed between 11:55 to 12:15 CET as the upper limit of the size mode increases from 1.5 to ~10 µm.

   In Fig. 4b, as ice crystals begin to grow in the chamber at higher $SS_w$, the mean depolarization ratio becomes more uniform, with a range of ~ 0 to 0.22 before 10:45 to a range of ~ 0.09 to 0.12 after 10:45.

Then at 11:55 CET (at 4 % $SS_w$) water droplet breakthrough initiates and the mean depolarization ratio decreases to about zero, consistent with the theoretical depolarization ratio of water droplets. These results show that the mean depolarization ratio of particles larger than 2 µm has a strong dependence on whether or not WDBT is occurring in the CFDC. This makes the mean depolarization ratio a useful tool for confirmation of the onset of water droplet breakthrough.

### 3.5 Optical properties of particles present in the CFDC

   In this section, the frequency distribution of depolarization ratios of particle populations present in the CFDC are investigated for comparison to the training datasets. First, all data from the FIN-02 campaign

was classified as WDBT conditions or normal operating conditions. Then particle diameters were used to determine the particle type. Aerosol particles during the FIN-02 campaign were generally < 2 µm in size. Since water droplets can bias this population during WDBT conditions, only those particles that are < 2 µm in diameter during normal operating conditions are defined as aerosols. Particles ≥ 2 µm in





diameter during normal operating conditions are identified as ice crystals. A third population is defined as "WDBT particles" and consists of particles ≥ 2 μm in diameter during WDBT conditions. This population typically consists of mostly water droplets, but can also include ice crystals. These three populations are referred to as "CFDC populations" in this manuscript.

Fig. 5 shows the depolarization ratio distributions of the CFDC populations interpreted to be ice crystals, water droplets, and aerosols. For the analysis completed to produce Fig. 5, 19 normal operating condition periods and 17 WDBT periods with variable time lengths were classified. Ice crystals achieve higher depolarization ratios than water droplets and aerosol. 13.2 % of ice crystals in the CFDC achieve a depolarization ratio > 0.3, compared to 1.5 % percent of water droplets and 0.3 % of aerosols. These

values are very similar to the percentages of training data particles that achieve a depolarization ratio > 0.3. Ice crystals achieve depolarization ratios > 0.3 more than 10 times more frequently than aerosol or water droplets. One interesting feature in the CFDC observations are the two Snomax® cases (cases 13 and 14 in Table 1 at -33 °C and -21 °C respectively) in Fig. 5. More particles with high depolarization ratios were observed than during the other 15 WDBT cases.  These particles are most likely ice crystals.

Since Snomax® bacteria are a particularly active INP it is not surprising that ice crystals dominate the population of particles in the CFDC even during WDBT (Wex et al., 2015), particularly at colder temperatures of these runs.

**3.6 Comparing CASPOL observations to model calculations**

      In this section, modeled and observed particles discussed in the preceding results section are compared. Fig. 6 shows modeled and observed depolarization ratios of particles as a function of diameter. The modeled results (green) are shown with the same shape conventions as Fig. 2. Observed results include

training (blue shapes) and CFDC (red shapes) ice crystals (pentagrams), aerosols (squares), and droplets/WDBT particles (circles). Observed values are accompanied by error bars representing the standard deviation of depolarization ratios of particles at the respective diameters plotted. The CFDC populations presented here include particles sampled from all FIN-02 experiments. The same





conventions are used here to process these particles: CFDC ice crystals are those larger than 2 u μm sampled under normal operating conditions, CFDC aerosols are those smaller than 2 μm sampled under normal operating conditions, and CFDC WDBT particles are those larger than 2 μm sampled under WDBT conditions.

In Fig. 6, both the model calculations and the observed results indicate that ice crystals have higher depolarization ratios than water droplets and aerosols on average at diameters above 5 μm. However, error bars show that the standard deviations of depolarization ratios at these sizes are very large and that the mean depolarization ratios of the observed particles displayed are not statistically significant from each other. This represents a major challenge in designing a new analysis method that uses
depolarization ratio to quantify INP.

     In section 3.5, the complex WDBT population was discussed. WDBT particles consist of both water droplets and ice crystals. Diffusional growth theory dictates that ice crystals will grow to larger sizes in the CFDC than water droplets (Pruppacher and Klett, 2012). Fig. 6 shows an increase in the depolarization ratio from ~ 0 to 0.25 in the CFDC WDBT region starting at ~6 μm. At diameters > 10
μm the mean depolarization ratio of WDBT particles is greater than or equal to the depolarization of CFDC ice crystals and training dataset ice crystals suggesting that these large particles are mostly or all ice crystals. It is inferred that particles in the 6 to 10 μm range are a mixture of water droplets and ice crystals.

     There are significant differences between modeled particles and their observed counterparts.
Observations show water droplets depolarizing light, but the observed mean depolarization ratio of water droplets is almost zero ($\delta \leq 0.05$). Another significant difference is that for both ice crystals and aerosols, the mean observed depolarization ratios are approximately 30% lower than the modeled depolarization ratio. One possible reason for the discrepancies between the model and observations is that the CASPOL depolarization detector underestimates the depolarization of particles due to the weak
depolarization of particles and relatively high detection limit of the CASPOL polarization detector. Another possibility is that the idealized model particles do not accurately depict the shape, composition, or other microphysical properties of the observed particles. Smith et al. (2016) found that after an ice crystal has nucleated, the geometry of the ice crystal can be modified leading to drastic differences in





the observed depolarization ratio. To investigate this, Smith et al. (2016) operated the Manchester Ice
Cloud Chamber at different temperatures and supersaturations to produce an assortment of ice crystal
morphologies including solid and hollow columns, plates, sectored plates and dendrites. Smith et al.
(2016) also compared observed and modeled depolarization ratio results and found that on average the
difference between modeled and observed depolarization ratios was ~120%. The CFDC results reported
in Fig. 6 include data from all of the runs sampled during FIN-02. The data set of the campaign
represents ice nucleation events over a broad range of temperature (-15 °C to -35 °C) and
supersaturation (0 % to 40 % $SS_i$) conditions. Thus, many different habits of ice crystals likely formed
in the CFDC, in part, contributing to the wide range of depolarization ratios reported in Fig. 6. Nicolet
et al. (2007) reported modeling results of single-particles that confirm that a wide range of
depolarization ratios can be detected for a single shape depending on the orientation. Non-preferential
orientation of particles in the CFDC is likey to contribute to the breadth of depolarization ratios
detected.

The observations are qualitatively consistent with the model in that ice crystals depolarize more light
than water droplets and aerosols. However, the discrepancies between the observed and modeled mean
depolarization ratios and the wide distributions of observed depolarization ratios dictate that we cannot
rely on a mean modeled depolarization ratio to identify and quantify ice crystals in the CFDC. Rather
than designing a theoretical model based on model calculations, we move forward by designing an
empirical model based on the CASPOL observed signals.

### 3.7 Designing an empirical model to quantify INP with depolarization ratio

The results in Sections 3.2 and 3.3 show that counting ice crystals in the CFDC using depolarization
ratio can be challenging since only ~13.5 % of ice crystals achieve a depolarization ratio that is greater
than 0.3 (Fig.s 3 and 5). This depolarization ratio threshold is a favorable criterion for ice crystals
because < 1% water droplets and aerosols achieve this depolarization ratio. However, during water
droplet breakthrough conditions, the water droplet concentration may be $10^3$ times greater than the ice
crystal concentration in the CFDC effectively reducing the signal to noise ratio ~1:1 or worse. To
combat these challenges, we use a linear regression fit derived from a simulated dataset designed with





the training data populations described in Sections 2.7 and 3.1. Other work has used a linear regression fit in a similar way to measure PM$_{2.5}$ with a ceilometer backscatter signal (Li et al., 2016).

To design the linear regression model, we create a simulated dataset from the training data populations. This simulated dataset has several segments with variable but known quantities of ice
crystals, water droplets, and aerosols. Second, using the simulated dataset, we optimize the choice of depolarization ratio threshold to maximize retention of ice crystals and removal of water droplet and aerosols. Next, we evaluate how well the optimal threshold ($\delta = 0.3$) performs in comparison to other thresholds over a large range of droplet and aerosol concentrations. Finally, we determine a linear regression fit that relates the number of nominally depolarizing particles to the known number of ice
crystals in a population determined by the number of ice crystals added from the ice crystal training dataset.

Each simulated dataset is created with 120 segments, with numbers of ice crystals in each segment ranging from 0 to 350. Fifty simulated datasets are generated with these identical ice crystal segments, but with incrementally increased droplet and aerosol concentrations. The ratio of water droplets and
aerosols is held constant across the 120 segments in a simulated dataset. The quantity of aerosols and water droplets in each simulated dataset is dictated by the multiplication factor $M$, where the number of water droplets = 100$M$ and the number of aerosols = 300$M$. $M$, increases by 1 for each iteration of the simulated dataset, ranging from 1 to 50, resulting in ranges of numbers of water droplets and aerosols of 100 to 5,000 and 300 to 15,000 respectively. The droplet, aerosol, and ice crystal
training datasets are randomized in time before the particles are selected from each population to form the simulated dataset. After these 50 simulated datasets are generated, the number of particles greater than or equal to a selected depolarization ratio threshold (ranging from 0 to 0.75 in increments of 0.05) and that are larger than 2 µm is determined for each of the 120 segments in the simulated dataset. A linear fit is determined for the relationship between the known ice crystal concentration and the number
of particles detected greater than or equal to the depolarization ratio threshold for only the simulated datasets generated where $M = 1$. The linear regression fit determined is then applied to all of the simulated datasets over the entire range of $M$. Only one fit is determined for each threshold because we cannot feasibly design a model that adapts to water droplet and aerosol concentration in the CFDC. An



$R^2$ value is determined to assess the goodness of the linear regression fit over all of the simulated datasets.

Fig. 7 shows the $R^2$ values as a function of $M$ and depolarization ratio threshold for each of the simulated datasets. The Fig. shows that $R^2$ values are very high for cases where aerosol and droplet

concentrations are low and when the depolarization ratio threshold is low. However, as the concentration of droplets and aerosol increase, the $R^2$ value decreases. This is especially true for lower depolarization ratio thresholds that are more sensitive to increases in droplet and aerosols. An optimal choice for depolarization ratio threshold is defined as a threshold that retains relatively high $R^2$ values across the entire range of $M$. Fig. 7 shows that the 0.3 and 0.35 thresholds are both optimal choices.

However, aerosol and water droplet concentrations in CFDC experiments are typically in the range $1< M< 20$. The mean $R^2$ value in this range of $M$ for the 0.3 and 0.35 thresholds 0.71 and 0.7 respectively. Thus we confirm that the 0.3 depolarization ratio threshold is the most appropriate threshold. The linear regression for the 0.3 threshold is provided in Eq. 9,

$$N_{INP} = 6.11 \, N_\delta + 22.20 \qquad\qquad (9)$$

where $N_\delta$ is the number of particles that have a depolarization ratio greater than 0.3 and $N_{INP}$ is the

derived INP number. Eq. 9 will be applied to all CFDC-CASPOL data collected during the FIN-02 campaign and the accuracy of this model will be determined.

**3.8 Application of the new analysis method to CFDC data collected during FIN-02**

INP concentrations were obtained using both the depolarization ratio method (Eq. 9) and the traditional method on CFDC data collected during the FIN-02 campaign. Three representative CFDC runs of Snomax® at -15 °C and at -20 °C and Arizona Test Dust at -25 °C are shown in Fig. 8. Each scan starts in subsaturated conditions with respect to water. Supersaturation is gradually increased until ice nucleation initiates and then further increased until WDBT occurs (represented by the red symbols in

Fig. 8). The reported concentrations reveal that the traditional and depolarization ratio methods generally agree during "ice only" periods (blue symbols in Fig. 8). In most cases there is clear disagreement between concentrations in WDBT periods, for example in the cases of Snomax® at -15 °C





and Arizona Test Dust at -25 $^\circ$C. This is expected since the traditional concentration is sensitive to an increase in water droplets that grow larger than the size cut applied in WDBT conditions, where INP concentrations are usually not reported. An exception to this can be seen in Fig. 8b, the Snomax® at -20 $^\circ$C. The concentrations from the two methods remain in good agreement as the supersaturation is

increased into the WDBT period. In this case, the ice crystal concentration is dominating the population in WDBT. The evidence for this is the high concentration of ice crystals that from 13:15 CET as observed in the size distribution time series in center panel Fig. 8b.

Fig. 9 summarizes the mean concentrations obtained through the traditional and new method for all periods when the CFDC was operational during FIN-02. In total, 27 "ice only" periods and WDBT

cases are included. A description of the date and time, aerosol composition, and temperature of each case is detailed in Table 1. The error bars report the CASPOL uncertainty, which is 39%. Fig. 9 shows that in all but 4 cases out of 27 (cases 2, 7, 9, and 23), the mean concentration of the new analysis method is in agreement with traditional analysis method for the "ice only" periods. Fig. 9 also shows that only 9 out of 24 WDBT cases have statistical agreement between the new and traditional analysis

method. At the onset of WDBT, the impact of water droplets on the INP concentration determined by the 2 μm size cut may not be very large and the concentration may closely resemble the true INP concentration, but as the $SS_w$ is increased more water droplets will be incorrectly counted in the traditional INP concentration. This phenomenon gives rise to the large error bars reported in some of the WDBT cases. In general, the observations reported in Fig. 9 are consistent with the assertion that the

traditional method and new method are in agreement during the "ice only" periods, and that during WDBT the traditional method is elevated in response to large water droplets polluting the INP concentration while the depolarization ratio method remains accurate.

To summarize the comparison between our new method and the traditional method during the "ice only" periods, the INP concentrations determined using the traditional method vs. new method are

plotted in Fig. 10. Each point on the plot represents data for a 1-minute segment. The black line in Fig. 10 is a 1:1 line. Since the analysis used to generate Fig. 10 only uses data collected under normal operating conditions (not WDBT), the traditional concentration can be considered ground truth. The data closely follows the 1:1 line, confirming that the depolarization ratio can be used to reliably retrieve



an INP concentration when no or few water droplets/aerosols are larger than 2 μm. To asses the performance of the new method we use mean percent error defined here as:

$$MPE = \frac{New\ Concentrtaion - Traditonal\ Concentration}{Traditional\ Concentration} \times 100\% \tag{10}$$

The mean percent error of the method is dependent on the INP concentration. Due to the high detection limit of concentration for the CASPOL, the mean percent error of the new method is ±500% when the traditional concentration is between 0 and 50,000 L$^{-1}$. However, at higher concentrations the MPE is typically ± 50 % or less. Additionally, Fig. 10 shows that at concentrations in the range 0 to 3 × 10$^6$ L$^{-1}$, the new method typically undercounts INPs, but over counts INPs at higher concentrations (> 3

× 10$^6$ L$^{-1}$). The mean percent error for the new method for all concentrations is ± 32.1 %.

Based on Fig. 10, the new analysis method only provides an accurate result when INP concentrations are high, which is only achievable in laboratory settings. For this reason, the method is not suitable to be used in a field setting where concentrations typically range from 0.1 to 100 L$^{-1}$ (e.g. Mason et al., 2016; Jiang et al., 2015; DeMott et al., 2003). Nonetheless, the new method is considered an improvement for

use during water droplet breakthrough, when the traditional method cannot be used.

As a final test of the new method during water droplet breakthrough periods, a reliable measure of INP at higher supersaturation conditions (when the TAMU CFDC is experiencing WDBT) is needed. Due to design and flow rate differences, the Colorado State University (CSU) CFDC does not experience the onset of WDBT until higher supersaturations than the TAMU CFDC, up to 108% or

higher depending on temperature (DeMott et al., 2015). Fig. 11 shows the comparison of the TAMU CFDC's traditional (2 μm size cut) and new method INP concentrations and the CSU CFDC INP concentration, collected during the FIN-02 campaign. As above, results of INP percent activated are reported from three CFDC runs including Snomax® at -15 °C and -20 °C and Arizona test dust at -25 °C. Concentrations used to calculate the percent activation are average concentrations of samples in a 1%

range of SS$_w$ conditions in the CFDC. Because the CSU CFDC has a different detector than the TAMU CFDC, CSU reports an INP concentration using a nominal size cut of 3 μm. Though not previously





discussed in the manuscript, a 5 μm size cut has also been used to report an INP concentration for the TAMU CFDC and is used here to provide an upper estimate for the INP concentration. Large symbols show data collected under normal operating conditions. Small symbols show data collected during WDBT conditions in the TAMU CFDC. The CSU CFDC did not experience WDBT in the data

reported in Fig. 11. The traditional concentration from TAMU and CSU and the new method concentration all are in reasonable agreement during "ice only" conditions for all three cases. During WDBT, the TAMU traditional concentrations increase in response to the water droplets that grow larger than the size criteria (2 μm or 5 μm). However, the new method remains in agreement with the CSU concentration. Fig. 11b shows a special case mentioned in previous discussion that has high activation

of INP. In this case there is negligible difference in the concentrations reported during "ice only" and WDBT periods since most particles activated prior to the onset of WDBT. In conclusion, the new method accurately determines the INP concentration in the presence of water droplets and can thus extend the range of operating conditions of the TAMU CFDC.

**4. Conclusions**

This manuscript presents a new analysis method that uses depolarization ratio to quantify INP concentrations in the TAMU CFDC using single-particle depolarization measured by the CFDC's CASPOL detector. Ice crystal, droplet and aerosol training populations were used to build simulated

datasets with known concentrations of aerosols, droplets, and ice crystals. The simulated datasets were evaluated to determine an optimal depolarization ratio threshold of 0.3 observed by the CFDC-CASPOL above which all particles were classified as ice crystals. Next, a simple empirical model using a linear regression fit was trained on simulated CFDC dataset using the 0.3 threshold and applied to CFDC data collected during the FIN-02 campaign. Concentrations of INP determined by the new

analysis method agree reasonably well with the traditional method (ice detection by size-segregation) under normal operating temperatures and supersaturations (with no large water droplets present) with a mean percent error of $\pm 32.1$ %. However, at INP concentrations <50,000 L$^{-1}$, the mean percent error of the new method is > 500 % due to a high concentration detection limit of the CASPOL. While high INP





concentrations of $10^4$ to $10^6$ L$^{-1}$ can be generated in laboratory settings, typical ambient INP concentrations range from 0 to 100 L$^{-1}$. For this reason, the new CASPOL depolarization method is recommended for CFDC laboratory experiments. A comparison between the CSU CFDC INP concentration and TAMU CFDC INP concentration derived from the new analysis method show

5   agreement even under conditions in which the TAMU CFDC is experiencing WDBT and CSU is not experiencing WDBT. We conclude that the new method can be used to extend the range of operating conditions in the CFDC. However, under conditions encountered in field studies, the traditional method is still preferred analysis method for counting ice nucleating crystals with the TAMU CFDC.

10   **Acknowledgments**

The authors acknowledge primary support from the National Science Foundation, Grant # ECS-1309854. EJTL, KJS, and PJD acknowledge support from NSF Grant # AGS-1358495. The FIN-02 and FIN-03 campaigns were supported by NSF Grant # AGS-1339264, and by the U.S. Department of Energy's Atmospheric System Research, an Office of Science, Office of Biological and Environmental

15   Research program, under Grant No. DE-SC0014487. Special thanks to Drs. Daniel Cziczo and Ottmar Möhler for their roles in coordinating the FIN-02 and FIN-03 studies, and for all research teams involved in making those studies possible.



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




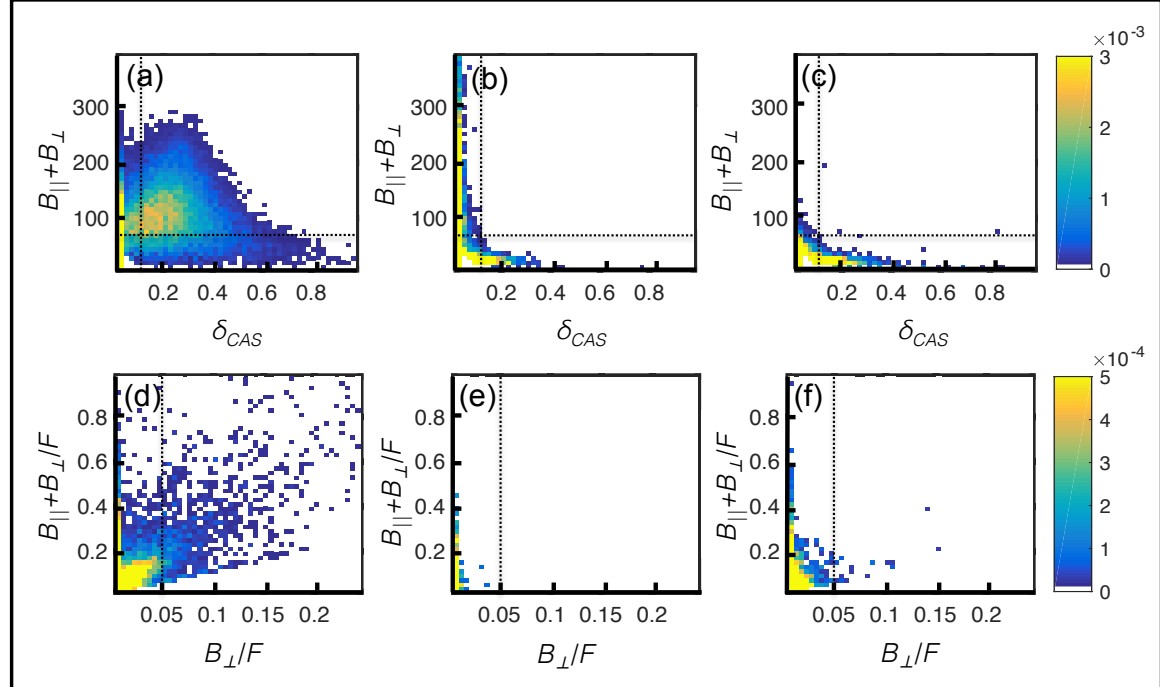

**Figure 1:** Optical signatures of training data populations: ice crystals (a, d), droplets (b, e), and aerosol (c, f). The CASPOL signals used to generate these signatures are parallel back scatter ($B_{\parallel}$), perpendicular back scatter ($B_{\perp}$), and forward scatter ($F$). The shading scales indicate the fraction of the training dataset that populates a grid cell.





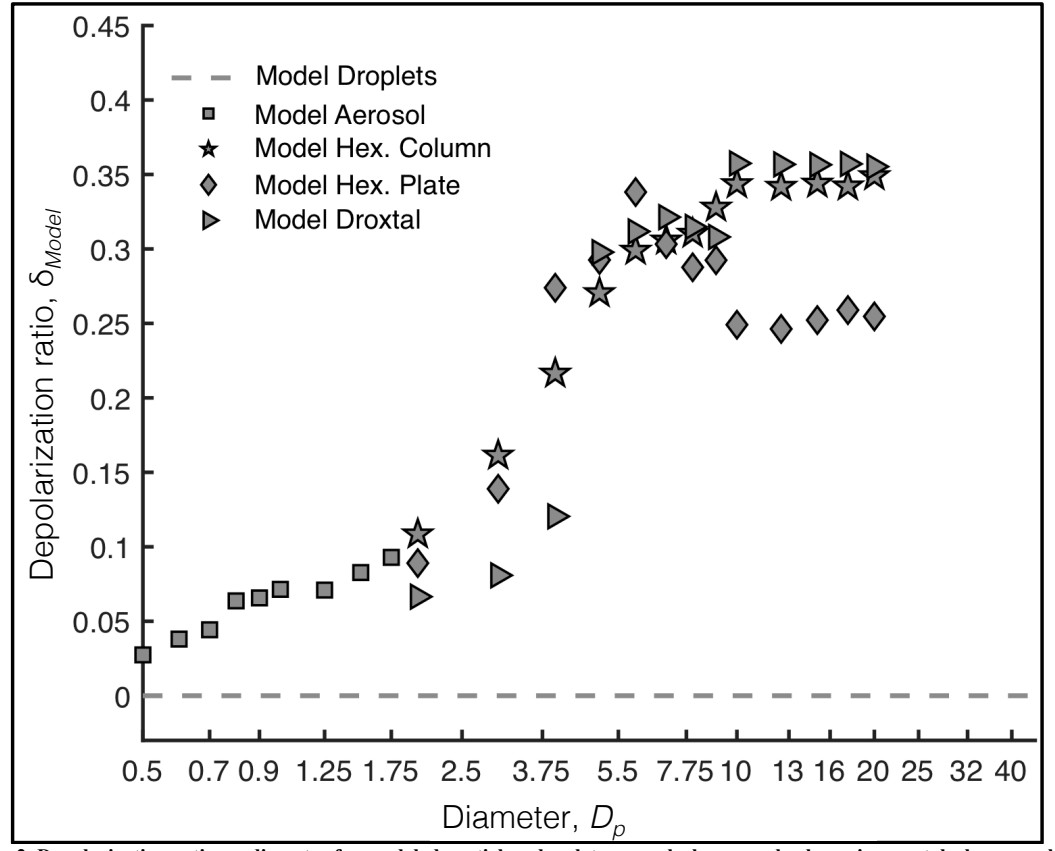

**Figure 2. Depolarization ratio vs. diameter for modeled particles: droplets aerosols, hexagonal column ice crystals, hexagonal plate ice crystals, and droxtals.**





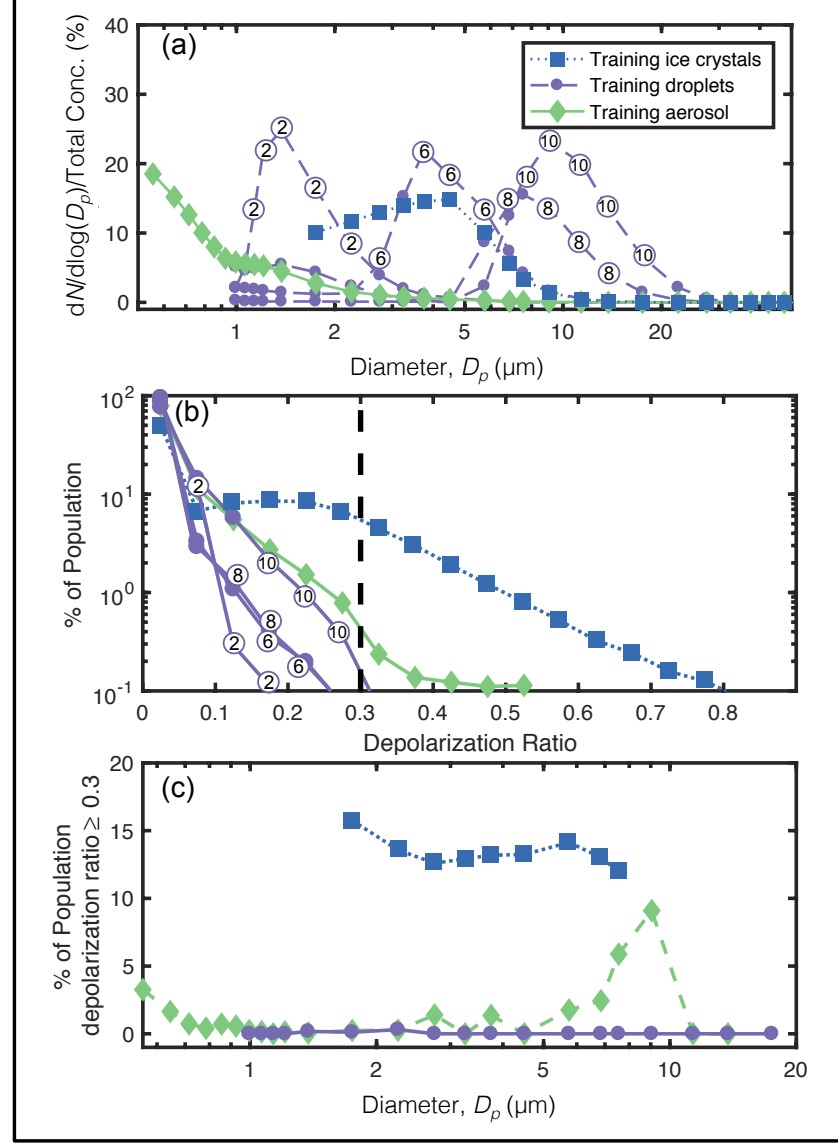

Figure 3: (a) Percent lognormal size distribution, (b) frequency distribution of depolarization ratios (c) the percentage of the particles with depolarization ratios above the threshold of 0.3 are shown for training data droplets, aerosols, ice crystals. In 1b, the depolarization ratio threshold value of 0.3 is indicated by the dashed line. In 1a and b, the numbers displayed in circles provide the diameter in μm of the VOAG data represented by that line.



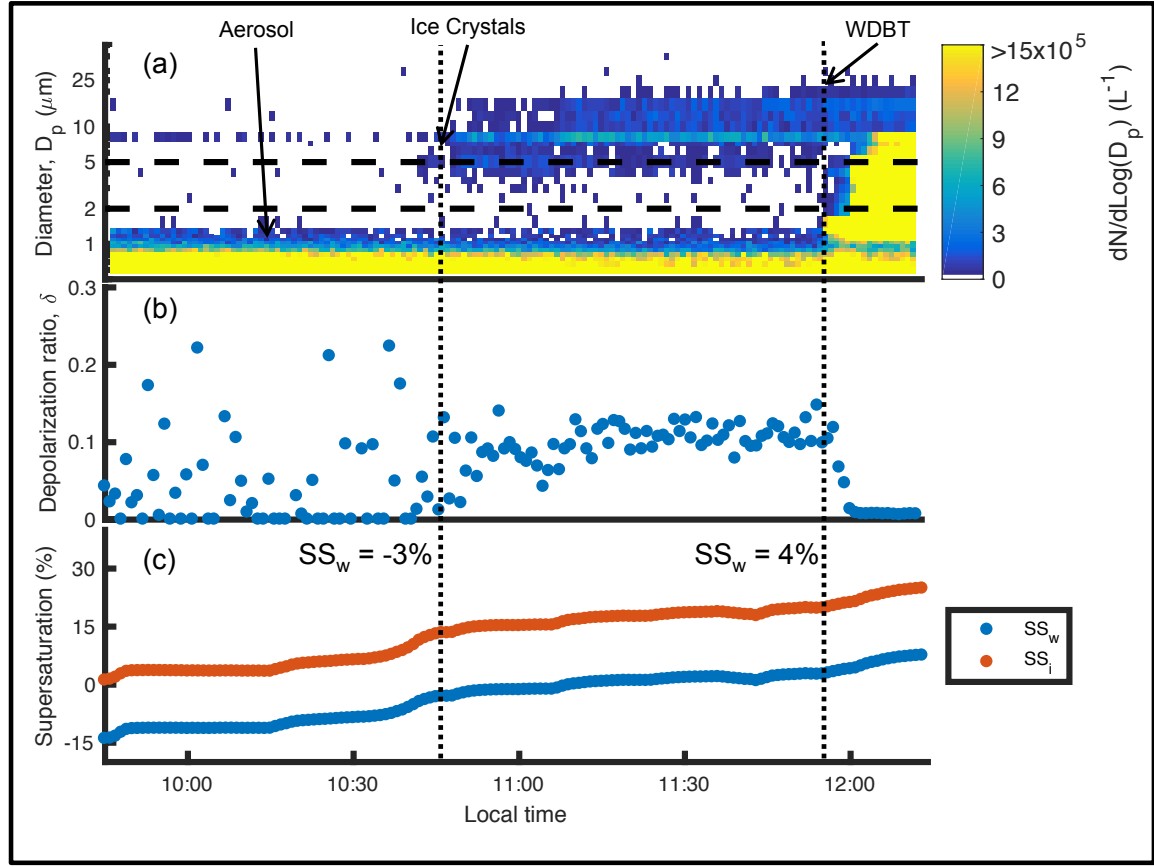

**Figure 4. (a) The normalized size distribution, (b) mean depolarization ratio of particles in CFDC with $D_p > 2$ µm, and (c) supersaturation conditions with respect to ice ($SS_i$) and water ($SS_w$) for a Snomax® scan on March 27 at -15 °C ± 1.5 °C.**





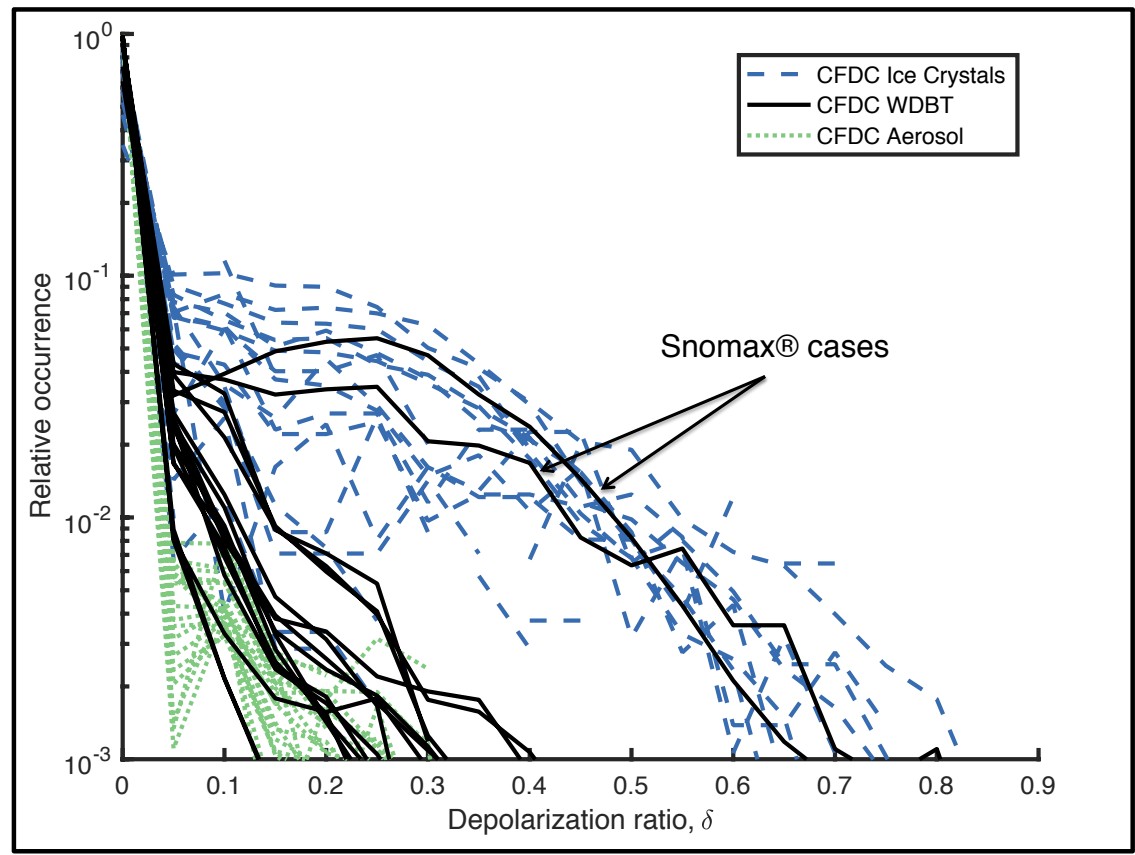

**Figure 5: Frequency distribution of depolarization ratios for CFDC populations: ice crystal periods (19 periods classified), WDBT periods (17 periods classified), and aerosol periods (19 periods classified). Mean temperatures of periods included range from -15 to -35 °C.**





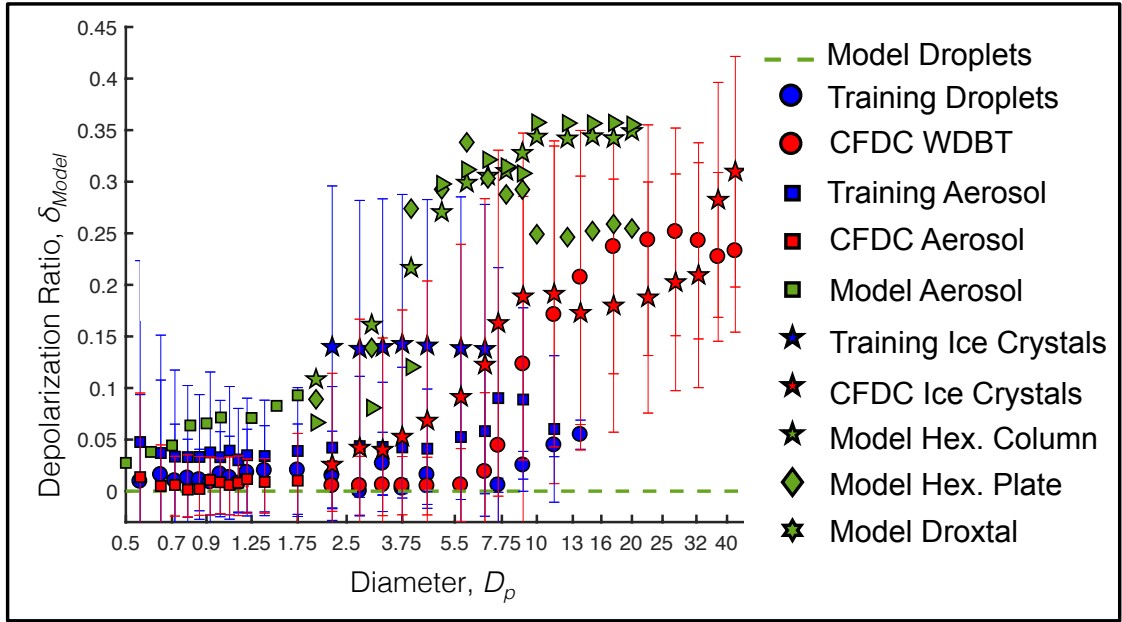

**Figure 6: Mean depolarization ratios vs. particle diameter for modeled and observed particles. Observed error bars provide a standard deviation on the depolarization ratios of particles at each reported size. No error bars are reported for model calculations.**



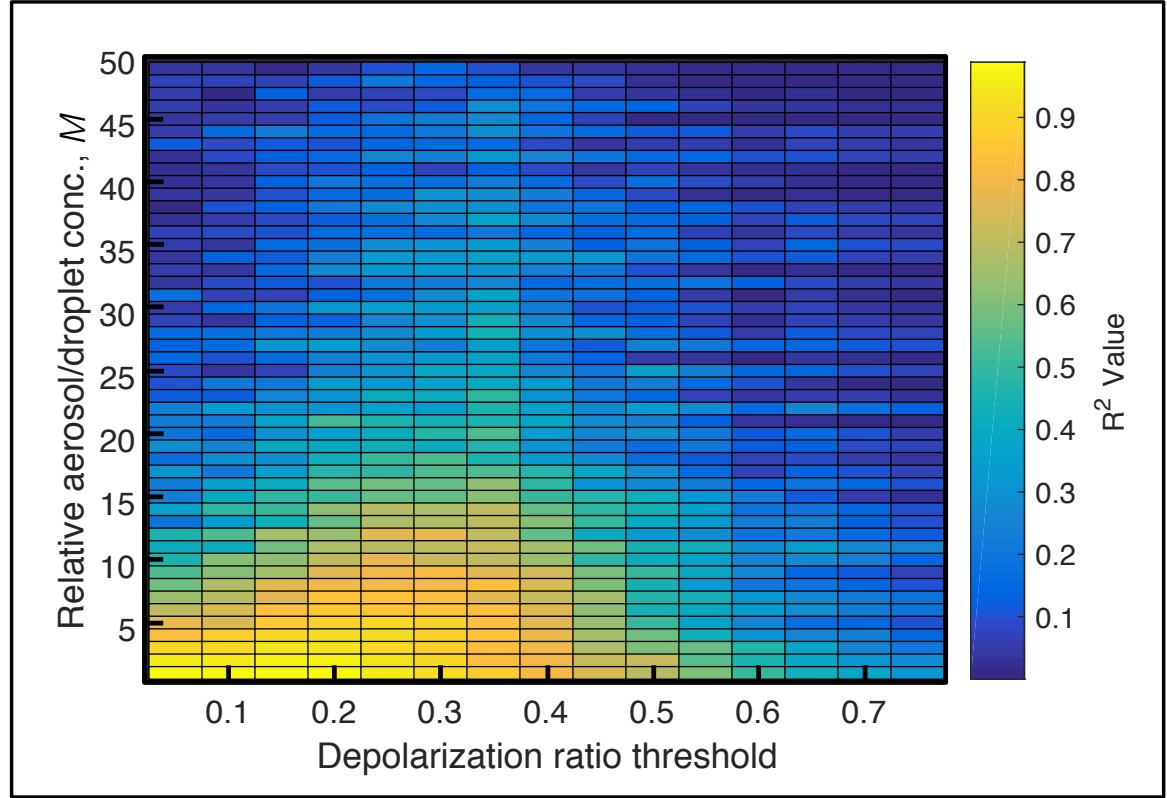

**Figure 7. $R^2$ values for linear regression fit as a function of depolarization ratio threshold for optimizing ice crystal differentiation and water droplet/aerosol concentration multiplication factor, $M$.**





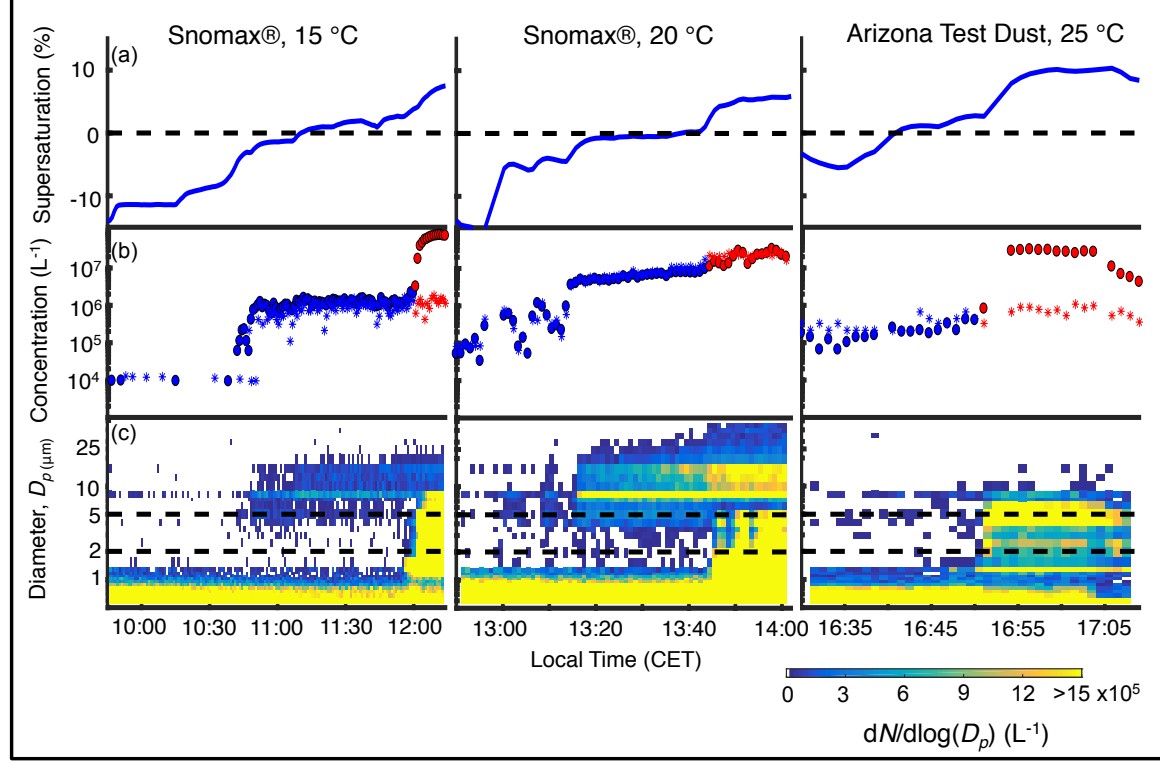

**Figure 8: Application of depolarization ratio method on three CFDC runs. Aerosol composition and temperature are labeled in the title. (a) Times series of supersaturation with respect to water,(b) INP concentrations as derived by the traditional method (L$^{-1}$) under normal operating conditions and WDBT conditions are reported as blue and red circles, respectively. INP concentrations as derived by the new depolarization ratio method are shown under normal operating conditions and WDBT conditions are reported as blue and red asterisks, respectively. (c) The normalized number distributions of all particles detected by the CASPOL. (c) the log-normal size distribution. Time is reported in local time (CET).**





**Figure 9:** Individual cases of "Ice Only" and "WDBT" INP concentration comparisons with the traditional size-cut and depolarization ratio methods. Error bars report the CFDC-CASPOL counting error of 39%.




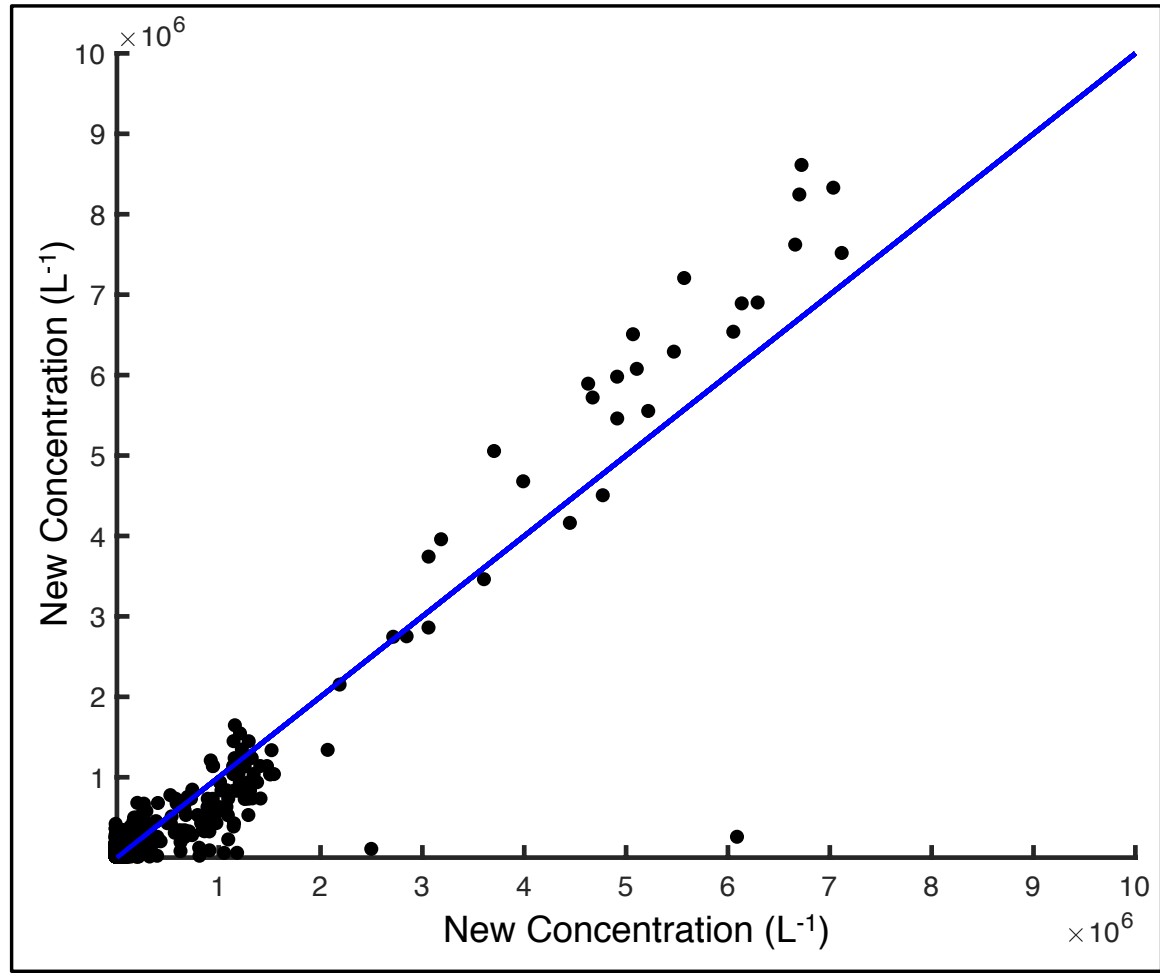

**Figure 10: Traditional INP concentration vs. new INP concentration with 1:1 line for "ice only" periods.**




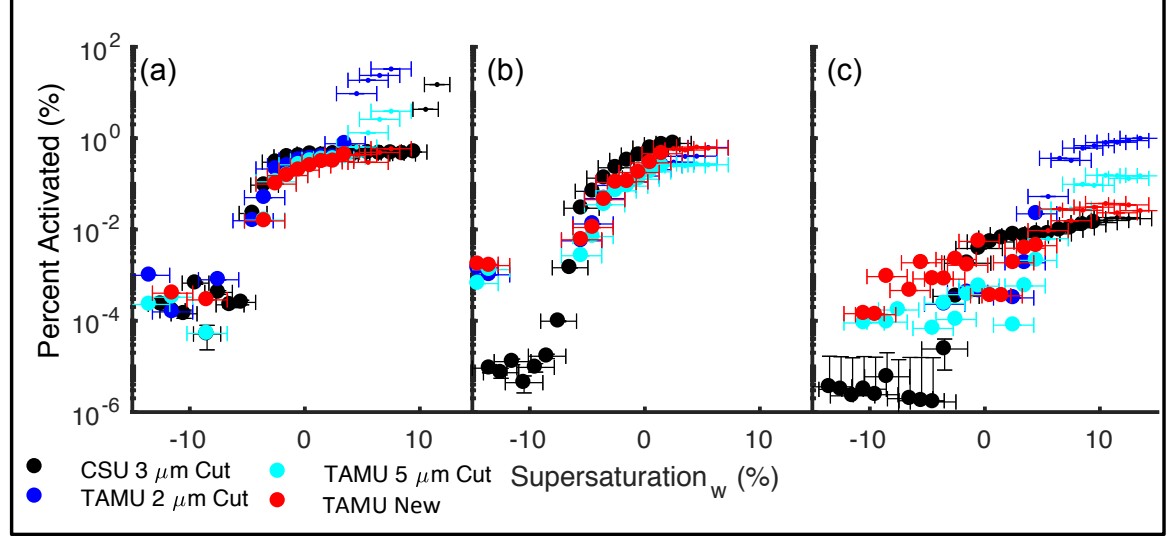

**Figure 11: TAMU CFDC versus CSU CFDC comparison. TAMU CFDC versus CSU CFDC comparison: Snomax® at -15 °C (a), Snomax® at -20 °C (b), and Arizona Test Dust at -25 °C (c). Small symbols indicate that those points were sampled in WDBT. TAMU 2 µm cut and 5 µm cut traditional INP fraction activated are shown in blue and cyan respectively. The TAMU new analysis method INP fraction activated is shown in red. The CSU 3 µm INP fraction activated is shown in black.**





| Case No. | Date Time | Composition | Chamber | Temperature (°C) |
|---|---|---|---|---|
| 1 | 3/24/15 10:13 | Arizona Test Dust* | AIDA | -25 |
| 2 | 3/24/15 11:25 | Arizona Test Dust* | AIDA | -20 |
| 3 | 3/24/15 12:48 | Arizona Test Dust* | APC | -19 |
| 4 | 3/24/15 16:02 | Argentinian Soil Dust* | AIDA | -19 |
| 5 | 3/24/15 17:29 | Argentinian Soil Dust* | AIDA | -18 |
| 6 | 3/24/15 18:28 | Argentinian Soil Dust* | AIDA | -24 |
| 7 | 3/25/15 10:15 | Argentinian Soil Dust* | AIDA | -25 |
| 8 | 3/25/15 11:22 | Argentinian Soil Dust* | AIDA | -28 |
| 9 | 3/25/15 12:35 | Argentinian Soil Dust* | APC | -28 |
| 10 | 3/25/15 16:48 | Arizona Test Dust* | AIDA | -25 |
| 11 | 3/25/15 17:51 | Arizona Test Dust* | AIDA | -28 |
| 12 | 3/19/15 17:45 | Arizona Test Dust | AIDA | -34 |
| 13 | 3/20/15 11:49 | Snomax® | APC | -33 |
| 14 | 3/20/15 13:28 | Snomax® | APC | -21 |
| 15 | 3/21/15 10:28 | Snomax® | AIDA | -16 |
| 16 | 3/21/15 11:12 | Snomax® | AIDA | -19 |
| 17 | 3/21/15 11:47 | Snomax® | AIDA | -20 |
| 18 | 3/21/15 12:54 | Snomax® | APC | -15 |
| 19 | 3/23/15 10:55 | K-Feldspar (Contaminated with Snomax®) | AIDA | -30 |
| 20 | 3/23/15 16:48 | K-Feldspar (Contaminated with Snomax®) | AIDA | -25 |
| 21 | 3/23/15 18:17 | K-Feldspar (Contaminated with Snomax®) | AIDA | -21 |
| 22 | 3/26/15 10:05 | Illite NX | AIDA | -25 |
| 23 | 3/26/15 11:09 | Illite NX | AIDA | -25 |
| 24 | 3/26/15 12:04 | Illite NX | AIDA | -28 |
| 25 | 3/26/15 12:44 | Illite NX | AIDA | -30 |
| 26 | 3/26/15 16:39 | Desert Dust | APC | -29 |
| 27 | 3/27/15 10:59 | Snomax® | APC | -16 |

Table 1 Date and time (CET), the composition of aerosol sampled, and the CFDC operating temperature (± 1.5 °C).
*Data collected during "blind tests" Sample composition was provided by the referees after the experiment was completed.