# Peer review of "Using depolarization to quantify ice nucleating particle concentrations: a new method"

_Atmospheric Measurement Techniques, 2017_

## Referee Comment (RC1) · Anonymous Referee #2 · 9 Aug 2017

The paper reports a new method, based on depolarization ratio, to enhance the calculation of ice nuclei concentrations in the occurrence of water droplet breakthrough. The method seems to be specific to the CFDC at Texas A&M University and to be applicable only in laboratory settings. For this reason, the wide applicability of the method might be limited. Despite that though, the issue to solve is an important one, especially considering the large uncertainties in the field of ice nucleation research. In addition, the technical work done for this comparison is considerable and involved also a modeling aspect. Therefore, I think the paper should be published. Overall the approach seems sound and well developed. Some clarification would be helpful in some instances, but overall the paper is well written. Some specific, rather minor comments:
1. Maybe I missed it, but I do not recall seeing mention of the specifications of the light

source in the CASPOL (wavelength, polarization, source, e.g. laser etc.). 2. On page 5 the authors describe the APC chamber, how are clouds produced in it, also through adiabatic expansion? 3. It would help to have some more detail on what causes the water droplet breakthrough, in what conditions, why it happens at different conditions in different instrument etc. For example around page 8 or so. 4. On page 9 on the first line: "precisely" seems a bit too strong; also LIDARs will have some finite field of view. 5. Still on page 9, the authors mention oil as having a similar real part of the imaginary index of refraction. I would think oil might have a different imaginary part of the index of refraction, with respect to water (also depending on the wavelength of the CASPOL). Maybe this is completely negligible, but could the absorption make any difference in the measurements or numerical simulations? 6. Page 10, line 7: remove "both" 7. Page 11, line 9 to 12. This sentence is not very clear to me. 8. Referring to figure 1, it seems like the total backscatter signal should have parenthesis in the label of the y axis. 9. Page 13, line 12, why is 1.75 um an upper limit for the CFDC? Are there some data on this item, or some published values? 10. Page 13, lines 24-26 and related figures: maybe I missed it, but how were the size distributions measured? 11. Page 16, line 24, I think "large" should be "larger". 12. Page 17, line 3, it seems like a "different" is missing when discussing the "statistically significant..." 13. Page 18, line 7, "like" should be "likely" 14. I found the section 3.7 hard to read and to follow. I am not sure what to suggest. Maybe a schematic of the algorithm would help, but as is, for me, it is very difficult to follow.

---

## Referee Comment (RC2) · Anonymous Referee #1 · 9 Aug 2017

The comment was uploaded in the form of a supplement:
https://www.atmos-meas-tech-discuss.net/amt-2017-166/amt-2017-166-RC2-supplement.pdf

---

## Referee Comment (RC3) · Anonymous Referee #3 · 9 Aug 2017

The comment was uploaded in the form of a supplement:
https://www.atmos-meas-tech-discuss.net/amt-2017-166/amt-2017-166-RC3-supplement.pdf

---

## Referee Comment (RC4) · Anonymous Referee #4 · 9 Aug 2017

This manuscript (Using depolarization to quantify ice nucleating particle concentrations: a new method by Zenker et al.) capitalizes on the ability of the CASPOL detection method to capture the depolarization information from particles, droplets and ice particles in the TAMU CFDC and identify them under different operating conditions. The method may be applicable to other systems but each CFDC is unique. The manuscript includes the development of a new empirical analysis method, to quantify ice nucleating particle concentrations and presents a way to deal with especially the data obtained during water droplet breakthrough, which is difficult to interpret. I believe that the manuscript topic does fall into the scope of AMT. Generally, the paper is readable, the analysis is carefully done and discussion points seem to be well supported by data. There is a limitation for this method in that higher concentrations only obtainable

in a laboratory are applicable; the authors are upfront about this limitation.

There are some major considerations that, if addressed, could strengthen the paper:

The authors may want to consider strengthening the end of their introduction to describe in more details the trajectory of work presented in the paper. Such a road map is limited here and more details could be helpful. In the body, there is little text regarding the comparison but there is a lot of text with many details regarding the development of the empirical analysis, yet these seem equally weighted in the introduction.

In terms of the training data, the text notes that no droplets below 2 $\mu$m were studied and this is reflected in figure 3. However, figure 6 shows training droplets at 0.7 $\mu$m. This is confusing. Further, since this size is a cut-off point for the analysis, it might be helpful to include smaller particles generation or to explain how the data in figure 6 was observed.

There may be minor scientific issues associated with the depolarization theory (that section of the paper was difficult to follow and there seemed to me to be some confusion or missing information associated with representations of matrices, matrix elements and values and/or units). In particular, the section on page 12 surrounding equations 6-7 is especially confusing. The authors note that these equations deal with the amplitude matrix, but then their inclusion in the equation appears to be an element with only one index. Further, it would be helpful to explain this part of the model further. What do these relationships (eqn 6-7) represent? I see how they combine to create eqn 8 but why?

It would also potentially be helpful for the authors to further discuss the use of the T matrix model for dust (and ice)? A recent technical note (Koepke et al., ACP, 2015, 5947) may be helpful. Generally, the paper would be enhanced with some additional details, clarity or references (and/or possibly even information in the experimental section) associated with the model calculations.

Overall, there is a lack of consistency within the text and figures where attention to detail would help. This is true, especially with the ordering of the types of particles within the different sections and also within the figures and captions. Further axis labels should include units where possible. A specific example is that in Fig. 6, there are both model and experimental results displayed but the y axis includes the model label and the x axis is missing units. Some additional specifics are included below.

Specific comments:

Pg 9, line 24: e is missing from the

Pg 10, line 11: "both" is unnecessary and confusing

Pg 10, line 19-20: In final copy, watch for placement of the minus sign

Pg 11, line 6-7: placement of training

Pg 11, line 8: based on the figure, the authors mean total backscatter vs depolarization ratio. I'd also suggest reversing the order in the follow up sentence on lines 8-10.

Pg 12, line 7: k is in eqn 3, but omega and t are not present. Is the equation missing time dependence? Also, r is not defined until line 11.

Pg 12, equation 4: related to above, are both matrices and matrix elements included?

Pg 12, missing comma after $P_{ij}$

Pg 14, line 5: I think you mean Fig. 3a here.

Pg 14, lines 14-17: This is confusing, please clarify. In figure 3b, it seems the % of total population of all particles having a depolarization ratio of $\sim$0.2 is close to 100%. How do other ratios exist for the population? This also makes interpretation of values in the text confusing.

Fig 6: Does it make sense to include the error bar information in the caption to make the figure less busy? Or at least remove and caption some of it?

Pg 17, line 1: typo of added "u". Also here you switch from >< notation to larger than and smaller than.

Pg 18, line 12: typo likely

Pg 20, line 4: double check wording for how this figure is introduced and also in the caption to be consistent and correct

Pg 20, line 5: suggest figure or Fig. 7

Fig 8: Caption could be improved, especially repetition in description of panel c.

Pg 20, line 26, suggest: In 2 out of 3 cases shown. Alternatively, you may want to clearly state (as you do later) that 27 cases/periods were evaluated (see Fig 9).

Pg 21, line 7: center panel of Fig. 8c?

Pg 21, line 12: is data missing from fig 9 or is it just hard to see?

Fig 10: Which axis contains the data for the new method? I suspect the x, but am unsure due to confusion noted above. Please clarify and update axes. Would it make sense to fit this data to observe is there is a small bias in the new data?

Pg 22, paragraph beginning on line 5: I am confused about how the errors in two regions can be 500 and 50% but overall it's 32%. I believe this is averaged values for each region considered. Is this the best way to present the uncertainty? Also as a minor detail, spacing when reporting numbers is inconsistent here and somewhat throughout the document, which would probably be fixed upon typesetting.

Figure 11: Consistency with previous figures and also double check captioning.

Pg 23, line 9: Does Fig 11b warrant more of a discussion? Can a literature comparison be included?

---

## Author Comment (AC2) · 5 Oct 2017

The comment was uploaded in the form of a supplement:
https://www.atmos-meas-tech-discuss.net/amt-2017-166/amt-2017-166-AC2-supplement.pdf

---

## Author Response (AR2)

*Authors' response on* **"Using depolarization to quantify ice nucleating particle concentrations: a new method"** *by* **Jake Zenker et al.**

**The authors thank the 4 anonymous reviewers for the detailed comments, all published on Aug 9. 2017. In the response below, we address each of the suggestions of the 4 reviews.**

5 **Anonymous Referee #1

*Referee Comment:*

Review of "Using depolarization to quantify ice nucleating particle concentrations: a new method" by Zenker et al.

General Comment This manuscript introduces a new method to distinguish between ice particles, aerosol

10 particles, and liquid water droplets at the water droplet breakthrough (WDBT) line in a continuous flow diffusion chamber. The traditional method to determine the concentration of ice nucleating particles (i.e., particle size) is not accurate at the WDBT and therefore, the proposed method can be of high importance. The new proposed method agrees well with the traditional method before the WDBT and it improves the detection of INPs at and above this line. However, this new method cannot be applied to field

15 measurements given that the uncertainty is very high when low concentrations of INPs are present. Therefore, this new method is only valid for laboratory experiments where high concentrations of INPs are usually achieve. Although the scientific goals are interesting and the experiments/analysis were carefully performed, the presentation of the manuscript is not the best. There are too many typos, some parts are repeated along the manuscript, and there are key references missing. It would be nice if a senior

20 researcher from the team can proof-read the revised version. **The reviewer did not find a major point; however, the following minor comments need to be addressed before its publication in AMT.**

*Authors' response:* Thank you. As the reviewer notes, we have stated in the original text that this technique is best applied to laboratory measurements due to signal to noise. Nevertheless, we agree with

25 the reviewer that developing such a method is of high importance due to the need for such a method, and we thank the reviewer for supporting this work.

*Authors' changes in manuscript:* The manuscript has been carefully revised and the minor comments of the reviewer are addressed item by item below.

**Minor Comments**

5  *Referee Comment:*

P2 L6: "depositional freezing" is incorrect given that "freezing" refers to the transition from liquid to solid. In deposition ice nucleation the liquid phase is not present.

*Authors' response:* Corrected.

10  *Referee Comment:* P2 L9-10: In all clouds or Mixed-phase clouds only?

*Authors' response:* In general, immersion freezing is the dominant nucleation mechanism for producing ice crystals in all clouds containing ice.

*Referee Comment:* P2 L17: Why mixed-phase clouds exclusively? Heterogeneous ice nucleation can also

15  takes place in cirrus clouds, for example.

*Authors' response:* We did not mean to imply that ice nucleation mechanisms are only relevant to mixed phase clouds. Mixed phase clouds are mentioned here because the current study addresses the specific challenges of accurate detection of ice in the presence of droplets.

20  *Referee Comment:* P2 L18: Add references after "GCMs". P2 L20: There are many studies showing this. I will rather cite a review paper instead.

*Authors' response:* Added "(e.g. Tan et al., 2016; Pithan et al., 2014)".

*Referee Comment:* P2 L22: Atkinson et al. (2013) and Yakobi-Hancock et al. (2013) are not field studies.

25  *Authors' response:* True. We have changed "Field measurements" to "Measurements".

*Referee Comment:* P2 L24-25: Other groups working on ice nucleation (besides the two cited here) have done a significant contribution as well. It would be better to divide the references by aerosol type. The recent reviews by Coluzza et al. (2017) and Kanji et al. (2017) nicely fit here.

*Authors' response:* This is certainly true. We were focusing on the TAMU CFDC which is the topic of

5 the manuscript and the CSU CFDC to which is it compared in this manuscript. But, we agree that other importance contributions should be mentioned.

*Authors' changes in manuscript.* We now include Coluzza et al. (2017) and Kanji et al. (2017).

*Referee Comment:* P3 L8: Add references after "crystals".

10 *Authors' response:* Reference to Bohren and Huffman, 1983 added.

*Referee Comment:* P3 L16: The Cziczo et al. (2017) review could be cited here.

*Authors' response:* Done.

15 *Referee Comment:* P6 L13: What do the authors mean with "processing chamber"?

*Authors' response*: Please refer to pg. 6 ln 4 which now reads "The aerosols then enter the CFDC processing chamber where temperature and supersaturation are controlled."

*Referee Comment:* P6 L14: Remove "TAMU". It was previously mentioned that CFDC will refer to the

20 TAMU CFDC.

*Authors' Comment:* Done.

*Referee Comment:* P7 L4: Why were the flows changed? Should it not be constant?

*Authors' response*: Flow adjustments were made to maintain conditions within the critical flow regime,

25 i.e. to ensue laminar flow, and avoid buoyancy effects under all operating temperature and supersaturations. Ideally, these flows would be constant; however in order to obtain the certain high supersaturation targets which were of interest during FIN02 evaluation of WDBT, adjustments were made.

*Referee Comment:*  P7 L4: Please add the uncertainty for 2 L min-1.

*Authors' response*:  Note that experimental uncertainty is not discussed in this sentence.  This is variation
in selected operating flows ($\pm$0.5 L min$^{-1}$) during FIN02. In contrast, uncertainty in sample flow is $\pm$0.1
L min$^{-1}$ based on the experimental precision in the total and sheath mass flow controlled.
*Authors' changes in manuscript:*  Added "$\pm$0.5 L min$^{-1}$."

*Referee Comment:*  P7 L4: I found the 1.5°C value quite high. Other CFDC report much lower values.
What is the reason for this?
*Authors' response*:  This is not an instrumental uncertainty, per se. It is a choice of data processing.  All
CFDC data collected at a mean temperature of X $\pm$1.5°C was included in the data set for a chosen
temperature X.

*Referee Comment:*  P7 L10: Remove "and" after "pressure".
*Authors' response*:  Done.

*Referee Comment:*  P7 L15-16: "The concentration of particles measured while the filter is in place is
subtracted from the total concentration measured by the CASPOL." Both are measured by the CASPOL.
I think it would be better to say: Total concentration measured during the supersaturation scan. To account
for this, a filter is placed ahead of the sample inlet in order to determine background signal of the CFDC
chamber. The background period that is closest to a given 1-minute sample period is then applied by
subtracting that background concentration from the total concentration measured by the CASPOL at the
sample time.
*Authors' response*:  We don't see exactly how the reviewer's suggestion would work, but we have revised
the sentence for clarity.
Authors' changes in the text: The text now reads, "… a filter is placed ahead of the sample inlet in order
to determine background signal of the CFDC chamber. The background period that is closest to a given

1-minute sample period is then applied by subtracting that background concentration from the total concentration measured by the CASPOL at the sample time."

*Referee Comment:* P7 L18: Add references after "crystals".
5 *Authors' response*: Added Bohren and Huffman, 1983.

*Referee Comment:* P7 L28: "Any droplets that remain larger than the 2 μm size cut will be miscounted as ice". This is based on who?
*Authors' response*: We are not certain what the Referee means to ask here. We feel it is clear as stated
10 that when the size-cut is set to 2 μm, any particles, be they ice or other composition, which are larger 2 μm will be counted as ice.

*Referee Comment:* P8 L7: "discern". Between what?
*Authors' response*: Changed to "determine".
15
*Referee Comment:* P8 L8: What do the authors mean with positive and negative artifacts?
*Authors' response*: This was a mistake on our part, because only positive artifacts are possible. "Positive artifacts" mean water droplets breaking through are counted as members of the ice crystal population. "Negative artifacts" would mean ice particles not counted because one thinks they are water droplets, but
20 in practice there is no way for that to occur.
*Authors' changes in manuscript:* "if the instrument is unintentionally operated at supersaturations above WDBT, droplets will be miscounted as ice crystals."

*Referee Comment:* P8 L11: This sound a bit awkward.
25 *Authors' response*: The original sentence was: "For the traditional analysis method to be successful, sample aerosols must not be larger than the applied size cut or they too will be miscounted as an INP."
*Authors' changes in manuscript:* Revised to: "In the traditional analysis, any sample aerosols larger than the applied size cut will also be miscounted as INPs."

*Referee Comment:* P8 L15: Add references after "signal".

*Authors' response*: This is based on empirical testing. There is no literature reference necessary.

5 *Referee Comment:* P8 L15: This is the fourth time the word "new" is used.

*Authors' response*: The topic of this manuscript is to compare one method, the "new method" developed here to the "traditional method", so we need to reserve the right to say "new" many times.

*Referee Comment:* P8 L16: What is "high" and "low"?

10 *Authors' response*: In the revised version of the document this sentence has been deleted due to other suggested revisions.

*Referee Comment:* P8 L16: Replace "our" with "the".

*Authors' response*: Removed "our" in the text.

*Referee Comment:* P9 L4: Add reference after "infinite"

*Authors' response*: This is a straightforward mathematical interpretation of Equation 2 above it. We do not feel it needs a reference.

20 *Referee Comment:* P9 L11: Delete "Using" after "…droplets."

*Authors' response*: Done.

*Referee Comment:* P9 L15: Add references for the 1.33 value.

*Authors' response*: Added Zajak and Hecht, 2002.

*Referee Comment:* P9 L16: Add the uncertainty for the droplet sizes.

*Authors' response*: Added.

*Author's change in the manuscript:* "As reported in Glen and Brooks (2013), the uncertainty in sizing due to differences in the complex refractive indices of oil and water are up to 30% based on a comparison of VOAG oil droplet calibrations of CASPOL to water-based calibrations performed by the manufacturer. For this project, droplets were generated with the diameters of $2 \pm 0.6$ μm, $6 \pm 1.8$ μm, $8 \pm 2.4$ μm, and
5   $10 \pm 1.5$ μm."

*Referee Comment:* P9 L18: Remove "and" after "frequency".
*Authors' response*: Here the grammar is correct as written.

10  *Referee Comment:* P9 L22: "sample flow is split between flow to the CASPOL" sound a bit awkward. Remove one "the".
*Authors' response*: Done.

*Referee Comment:* P9 L22: Remove "and" after "controller".
15  *Authors' response*: Here the grammar is correct as written.

*Referee Comment:* P9 L24: Replacer "are" with "were".
*Authors' response*: Done.

20  *Referee Comment:* P10 L4: "in aerosols"?
*Authors' response*: Changed to "of aerosols"

*Referee Comment:* P10 L12-23: "in the absence of activated liquid droplets". Do the authors mean in the absence of INPs?
25  *Authors' response*: No. The referee is correct that homogenous freezing occurs without INP. However, the point we're making here is that the experiment under cold, dry temperatures to reduce the chance of any unwanted droplet formation.

*Referee Comment:* P10 L15: "-11 ± 1.5 % SSw"? Something is wrong here.

*Authors' response*: We have rearranged and simplified the sentence for clarify.

*Authors' changes in manuscript:* The text now reads, "the CFDC was operated at −55 ± 0.2 °C and 51 ± 2.3 % $SS_i$(-11 ± 1.5 % $SS_w$)..."

*Referee Comment:* P10 L26: Remove "TAMU". See comment on P6

*Authors' response*: Done.

*Referee Comment:* L14 P11 L16: "Fig.s 1" should be "Fig. 1".

10  *Authors' response*: Fixed.

*Referee Comment:* P12 L17 and P13 L6-8: Why did the authors choose dust-like as the model for aerosol particles? How about biological particles? Soot?

*Authors' response*: Indeed, aerosols come in a wide variety of compositions. Performing scattering
15  calculations on all atmospheric aerosol types would be beyond the scope of this study. We chose dust as a relevant choice due to its widespread presence in the atmosphere and the known action of many dusts as INPs. Also, many of the organic and inorganic salt particles in the atmosphere will be in the form of solution droplets. Since those will have similar scattering properties to the water droplets, which were already included, the non-spherical dust also provides a good compliment to the spherical particles.

*Referee Comment:* P13 L27: Please indicate to what Figure the authors are referring to.

*Authors' response*: Fig. 1a is now stated.

*Referee Comment:* P14 L23-24 and along the manuscript: Please use "WDBT" instead of "water droplet
25  breakthrough". This was defined in P8 L1.

*Authors' response*: True, but we find it helpful to write it out in full one more time for readers who are likely to be unfamiliar with the term.

*Referee Comment:* P15 L5: Please indicate to what Figure the authors are referring to.

*Authors' response*: Fig 1a is now included.

*Referee Comment:* P16 L24-26: Replace "um" with "μm" to be consistent.

5 *Authors' response*: Corrected.

*Referee Comment:* P17 L8: I think the year of the Pruppacher and Klett book is incorrect.

*Authors' response*: Thank you, we have corrected this oversight.

10 *Referee Comment:* P17 L23-24: "the geometry of the ice crystal can be modified leading to drastic differences in the observed depolarization ratio." Can the authors report the time scale under which this is valid? i.e., how many seconds/minutes are needed for an ice crystal to change its geometry?

*Authors' response*: This comment pertains to the study of Smith, 2016. . They grew ice crystals in the Manchester Ice Chamber which is similar in size and design to the AIDA chamber. They operated the

15 chamber at various temperatures and grew ice crystals in the chamber. The change in relative humidity reportedly causes the evolution of ice crystal geometry. To monitor the changes, sample were taken every minute over 5-6 minutes. There were visually noticeable changes in to the habit at each minute increment (e.g. hollow hexagonal columns evolve slowly into solid hexagonal columns, dendrites evolve into hexagonal plates). The paper does not report a specific timescale that is necessary for a detectable change

20 in ice crystal shape to occur, but their results suggest that changes may occur rather rapidly with changing conditions in the chamber (over a minute or less). While this is an interesting study, we feel it is beyond the scope of our study and interested readers are referred to the Smith paper.

*Authors' changes in the manuscript:* None.

25 *JAKE Referee Comment:* P17 L24 and 27: Add the year of the Smith et al. paper.

*Authors' response*: The year is now in the text "Smith et al. (2016)"

*Referee Comment:* P17 L28: "(2016)" is out of place.

*Authors' response*:  Corrected.

*Referee Comment:*  P18 L2: Add "field" before "campaign".

*Authors' response*:  FIN-02 was a laboratory campaign, FIN-03 was a field campaign. These are now correct everywhere in the text.

*Referee Comment:*  P19 L25: Please indicate to what Figure the authors are referring to.

*Authors' response*:  Fig. 7 is now mentioned in the text.

*Referee Comment:*  P22 L3-4: How about to include kanji et al. (2017)?

*Authors' response*:  Good idea. Done.

*Referee Comment:*  P22 L8: "the Colorado State University (CSU) CFDC". This was defined already in P3 L18.

*Authors' response*:  True, but in discussion with coauthors we decided a little repetition for clarify was helpful.

*Referee Comment:*  P23 L19: Add "only" after "experiments".

*Authors' response*:  Done.

*Referee Comment:*  P25-30: Be consistent with the journal names in the references. Either add the full name or their abbreviation.

*Authors' response*:  Abbreviations are now used in accordance with the AMT manuscript preparation guidance.

*Jake Referee Comment:* P25-30: The page numbers in several references are missing (e.g., DeMott et al. (2017), Levin et al. (2016), McCluskey et al. (2016), McFarquhar et al. (2011)).

*Authors' response*:  All page numbers are now included where applicable.

*Referee Comment:* P25-30: References need to be up to date.

*Authors' response:* We have added a few references from 2017 per the reviewer recommendations (Coluzza et al., 2017; Cziczo et al., 2017; Kanji et al., 2017).

Figure 2: Given that there is no extra-charge for colored-figures in AMT, I suggest to add color to this figure to improve its readability.

*Authors' response:* Since the symbols and colors on Fig. 2 were chosen to match the same data sets included on page. 6, we prefer to keep the original scheme.

*Referee Comment:* Figure 3: Blue circles in panel's b and c should be blue squares.

*Authors' response:* This has now been fixed.

*Referee Comment:* Figure 11: "TAMU CFDC versus CSU CFDC comparison."

15 Is written twice in the figure caption.

*Authors' response:* Corrected. Thank you.

*Referee Comment:* Table 1: Add ":" after "1" for consistency with the Figures.

*Authors' response:* Done.

References:

Coluzza, I., Creamean, J., Rossi, M. J., Wex, H., Alpert, P. A., Bianco, V., Y. Boose, C. Dellago, L. Felgitsch, J. Fröhlich-Nowoisky, H. Herrmann, S. Jungblut, Z.A. Kanji, G. Menzl, B. Moffett, C. Moritz, A. Mutzel, U. Pöschl, M. Schauperl, J. Scheel, E. Stopelli, F. Stratmann, H. Grothe, and D. Schmale III

25 (2017). Perspectives on the Future of Ice Nucleation Research: Research Needs and Unanswered Questions Identified from Two International Workshops. Atmosphere, 8(8), 138.

Cziczo, D. J., Ladino, L., Boose, Y., Kanji, Z. A., Kupiszewski, P., Lance, S., Mertes, S., and Wex, H. (2017). Measurements of Ice Nucleating Particles and Ice Residuals. Meteorological Monographs, 58, 8.1-8.13.

5   Kanji, Z.A., Ladino, L., Wex, H., Boose, Y., Burkert-Kohn, M., Cziczo, D.J., and Krämer, M. (2017). Overview of Ice Nucleating Particles. Meteorological Monographs, 58, 1.1-1.33

**Anonymous Referee #2

*Referee Comment:* The paper reports a new method, based on depolarization ratio, to enhance the calculation of ice nuclei concentrations in the occurrence of water droplet breakthrough. The method

5 seems to be specific to the CFDC at Texas A&M University and to be applicable only in laboratory settings. For this reason, the wide applicability of the method might be limited. Despite that though, the issue to solve is an important one, especially considering the large uncertainties in the field of ice nucleation research. In addition, the technical work done for this comparison is considerable and involved also a modeling aspect. Therefore, I think the paper should be published. Overall the approach seems

10 sound and well developed. Some clarification would be helpful in some instances, but overall the paper is well written.

*Authors' response.* Thank you for your positive review. We have revised the manuscript in order to provide clarification in the specific instances below.

*Referee Comment:* Some specific, rather minor comments:

1. Maybe I missed it, but I do not recall seeing mention of the specifications of the light source in the CASPOL (wavelength, polarization, source, e.g. laser etc.).

*Authors' response.* The light source is a linearly polarized 680 nm laser.

20 *Authors' changes in manuscript.* Added on pg 6 ln 20, "Laser light (680 nm) is scattered by single particles entering the CASPOL and detected by three detectors…"

*Referee Comment* 2: On page 5 the authors describe the APC chamber, how are clouds produced in it, also through adiabatic expansion?

25 *Authors' response:* No. To clarify, in this experiment, no clouds are produced in the APC chamber during the experiment. The APC is used to provide a uniform high concentration of aerosols generated by filling the APC chamber with aerosols produced by atomization and solid aerosol generation methods.   The ice cloud particles sampled by the CFDC-CASPOL are produced within the CFDC's processing chamber

under conditions of controlled saturation conditions produced by varying the temperature gradient between the inner and outer iced walls of the chamber.

*Authors' changes in manuscript:* On pg 4, ln 28, the text now reads, "The APC was used during FIN02 to provide a uniform high concentration of aerosols of various compositions, which were generated by filling the APC chamber with aerosols produced by atomization and solid aerosol generation methods and were subsequently distributed to the participating ice nucleation instruments"

*Referee Comment 3:* It would help to have some more detail on what causes the water droplet breakthrough, in what conditions, why it happens at different conditions in different instrument etc. For example around page 8 or so.

*Authors' response:* Thanks for this good suggestion. Water droplet breakthrough is the term used to describe the arrival of droplets reaching the detector of an ice nucleation chamber where they will be miscounted by most as ice particles by most detection methods. This arises when the chamber is operated under supersaturation conditions and supercooled droplets form in the initial sections of the processing chamber. Most CFDC designs include a section following growth chamber, referred to as an evaporation region. The evaporation region is maintained under conditions at which the Bergeron process is active, that is conditions, which are subsaturated with respect to droplets. Thus, droplets shrink or evaporate entirely while at the same time the conditions are supersaturated with respect to ice, allowing ice crystals to grow.

The specific conditions of the evaporation chamber vary from instrument to instrument. Another cause of differences in WDBT between instruments is the selection of the size cut-off for distinguishing INPs by size alone. For example, using the traditional strategy of relying on a nominal size-cutoff to define INP, if an operator chooses 2 microns as the diameter above which all particles are presumed to be ice, then a water droplet need only be 2 microns in diameter to "break through," whereas if the operator chooses a 5 micron size cut-off same detector operating under all the same conditions, only water droplets will necessarily have to grow to 5 microns to break through and be miscounted as ice. So, it is the combination of chamber dimensions, flow rates, operating conditions (temperature and supersaturation)

in the growth and evaporation regions, and choice of detector and size cut-off which collectively determine WDBT for a certain instrument.

*Authors' changes in manuscript:* Page 8 has been revised to include addition details, "WDBT is a common
5   issue in continuous flow ice nucleation instruments, although the point at which WDBT occurs varies between instruments of differing dimensions and even as a function of operating conditions (especially temperature) within a single instrument (Rogers et al., 2001, DeMott et al, 2015, Garimella et al., 2016). CFDCs in use today are custom-built instruments which vary in physical dimensions and choice of detector, although all operate under the same basic principles. Due to the combination of chamber
10   dimensions, flow rates, operating conditions (temperature and supersaturation) in the growth and evaporation regions within the instrument, and the choice of detector and size cut-off, WDBT varies from instrument to instrument.  In some cases, it can be difficult to determine when WDBT is occurring, and if the instrument is unintentionally operated at supersaturations above WDBT, droplets will be miscounted as ice crystals."

*Referee Comment* 4: On page 9 on the first line: "precisely" seems a bit too strong; also LIDARs will have some finite field of view.
*Authors' response:* We think our point is best made by keeping "precisely" here to emphasis the difference in backscattering angle of the lidar at 180° from the CASPOL backscatter at 168° to 176°.

*Referee Comment* 5: Still on page 9, the authors mention oil as having a similar real part of the imaginary index of refraction. I would think oil might have a different imaginary part of the index of refraction, with respect to water (also depending on the wavelength of the CASPOL). Maybe this is completely negligible, but could the absorption make any difference in the measurements or numerical simulations?
25   *Authors' response:* The uncertainty in sizing due to differences in the complex refractive indices of oil and water are up to 30% based on a comparison of VOAG oil droplet calibrations of CASPOL performed in our laboratory in comparison to the manufacturer's water-based CASPOL size calibrations (with the oil droplets being overestimated).   This is discussed in detail in our previous work (Glen and Brooks,

2013). Comparable uncertainties are expected for the simulations.

*Authors' changes in the manuscript:* Added "As reported in Glen and Brooks (2013), the uncertainty in sizing due to differences in the complex refractive indices of oil and water are up to 30% based on a comparison of VOAG oil droplet calibrations of CASPOL to water-based calibrations performed by the manufacturer."

*Referee Comment* 6: Page 10, line 7: remove "both"

*Referee Comment* 7: Page 11, line 9 to 12. This sentence is not very clear to me.

*Authors' response:* We have revised this text.

*Authors' changes in manuscript.* The text now reads, "Each training dataset contains some particles that are highly backscattering and some particles that are highly depolarizing, but only the ice crystal population contains particles that have both a high depolarization ratio and high backscatter signal."

*Referee Comment* 8: Referring to figure 1, it seems like the total backscatter signal should have parenthesis in the label of the y axis.

*Author Response:* This has been changed as suggested by the Referee.

*Referee Comment* 9: Page 13, line 12, why is 1.75 um an upper limit for the CFDC? Are there some data on this item, or some published values?

*Authors' response:* The CFDC's standard procedure is to operate with a cyclone impactor installed at the inlet which removes particles larger than 1.75 micron diameter. This was mentioned earlier but not on page 13.

*Authors' changes in manuscript.* This sentence has been removed because aerosol calculations for larger sizes are included to address a different Referee suggestion.

*Referee Comment* 10: Page 13, lines 24-26 and related figures: maybe I missed it, but how were the size distributions measured?

*Authors' response:* The forward scattering detector of the CASPOL detects particles on an individual basis and sorts those particles into a series of size bins ranging from 0.6 to 50 micrometers optical diameter.

*Authors' changes in manuscript.* Details above are now included on pg 7, ln 4.

*Referee Comment* 11: Page 16, line 24, I think "large" should be "larger".
*Authors' response:* Changed.

*Referee Comment* 12: Page 17, line 3, it seems like a "different" is missing when discussing the
10  "statistically significant..."
*Authors' response:* We agreed. Added.

*Referee Comment* 13: Page 18, line 7, "like" should be "likely"
*Authors' response:* Corrected.

*Referee Comment* 14: I found the section 3.7 hard to read and to follow. I am not sure what to suggest. Maybe a schematic of the algorithm would help, but as is, for me, it is very difficult to follow.

*Authors' response:* For clarity, we've structured the text and added details throughout the section to guide
20  the reader. Further, we've added a table that lays out how the training datasets are generated. There were specific points of confusion pointed out by other referees that we addressed here that strengthen the paragraph as well.

**Anonymous Referee #3

Review to "Using depolarization to quantify ice nucleating particle concentrations: a new
Method" by Zenker et al. AMTD, 2017    the manuscript by Zenker and coworkers describes the applica
5    tion of a new method to discriminate particle types in continuous flow diffusion
Chamber (CFDC) studies using a depolarization signal obtained from the Aerosol
Spectrometer with POLarization (CASPOL). The work is motivated by the difficulties faced
when particle type discrimination is purely based on particle size ("traditional method"), as particl
es of the same size do not necessarily need to be of the same type. Using a set of
10   training data, where only ice crystals, aerosol particles or (cloud) droplets exist the authors
show how the CASPOL depolarization signal can be used to differentiate these particles types
based on optical signatures.   A linear regression fit model is then used to optimize the
depolarization ratio value considered as (threshold) criterion to differentiate particle types,
concluding with an optimal value of 0.3. The corresponding linear regression is then used
15   to calculate the ice nucleating particle (INP) concentration for an extensive CFDC data
set and the results for the "new" and the "traditional" method are compared. I believe that
the topic of reliable particle type discrimination in CFDC studies (when operating under water
droplet breakthrough (WDBT) conditions) is inherently complex and needs to be addressed in
the future. This manuscript certainly provides motivation to do so and the presented results show
20   evidence that using depolarization ratio can contribute to a more accurate discrimination
of particle type in CFDC studies than is currently done by size discrimination and
ultimately leads to a better quantification of INP concentrations. The manuscript at the current state w
ould benefit from restructuring and major revisions to clarify certain key aspects of
the data analysis and interpretation. Once all concerns given in the following are properly
25   addressed, this manuscript may be suitable for publication in AMT.

*Authors' response:*   Thank you.  We agree that WDBT needs to be addressed for ice nucleation
measurements to be more accurate and performed reliably under a broader range of atmospherically

relevant conditions. Also we agree that depolarization ratio can improve our discrimination between INP and wayward droplets reaching the detector, as our manuscript shows for the laboratory experiments herein.

*Authors' changes in the text:* We have restructured the introduction, moved the section on particle depolarization, and significantly revised the data analysis and interpretation sections, as well as the modeling section, as discussed in the point by point response and in response below and in response to the other reviewers. We feel that the manuscript has improved greatly in readability and clarity and hope that the referee agrees.

**General Comments:**

*Referee*                                                                                  *Comment:*
Section 1: Please focus more on the core topic of the manuscript and provide background
for        the       particle discrimination in CFDC studies. I also encourage the authors to motivate the
need for a better                                     particle type/phase discrimination in order to more
clearly indicate the additional value obtained from new methods as presented in this
manuscript.

*Authors' response:* We have restructured the introduction, following the later comment of the reviewer that the benefit of the new methods does become clear on page 4. We have moved that text forward to earlier in the introduction. We do note that current phase discrimination studies available in the literature are cited, and there are not a very large number available, which is further reason that we'd like to see the present study published.

*Authors' changes in the manuscript:* Please see the revised introduction.

*Referee Comment:* Section 2: This section requires restructuring and currently misses important technical details for the instruments used (e.g. TAMU CFDC) or appropriate references. Please add mo re instrumental details to the manuscript.

*Authors' response:* Section 2 has been restructured, with subsections 2.1 and 2.2 reordered (switched), and has been revised according to specific Referee comments here and in the other reviewers. As cited in Section 2, the TAMU CFDC and CASPOL have been discussed in great detail in our previous work (Glen & Brooks, 2013 and 2014, and Glen, 2014.)

5   *Authors' changes in the manuscript:* Please see the revised section 2 in the text.

*Referee Comment:* In Section 3, the creation of the simulated data sets and implementation of the regression model  remains unclear to me. I struggled to follow how the optimal depolarization
ratio threshold is identical to the one presented in Section 3.3, which I assumed

10   at this stage to be empirical.

*Authors' response:* In section 3.3, the training data sets are introduced. In Figure 3B, the population of particles is plotted against depolarization ration.   It can be seen from this figure that droplets have depolarization ratios up to 0.3. Therefore, we visually assign 0.3 as the nominal depolarization threshold cut-off. At this point, the caveat remains that a small percentage of aerosols do have depolarizations

15   greater than this threshold.   However, in the case of aerosols, there are 2 lines of defense against any aerosols being accidently counted as ice nucleation. In addition to the majority of aerosols having in addition to the depolarization threshold, aerosols with sizes above 1.75 micron diameter are physically removed from the sample upstream of the CFDC chamber.

20    In section 3.7, the linear regression model is introduced and used to optimize the cut-off. The statistically significant results in confirm that 0.3 is an optimal choice of threshold.

*Authors' changes in the text:* For clarity, section 3.3 now includes the statement. "For each training data set, the frequency distribution of depolarization ratio reported as a percentage of the total particles in the data set is shown in Fig. 3b. It can be seen from this figure that droplets have depolarization ratios up to

25   0.3. Therefore, we visually assign 0.3 as the nominal depolarization threshold cut-off for differentiating between ice crystals and non-ice particles. Unfortunately, it can also be seen that a small percentage of aerosols do have depolarizations greater than this threshold.   However, since aerosols with sizes above 1.75 micron diameter are physically removed from the sample upstream of the CFDC chamber, the

combined consideration of size and depolarization may prove a robust strategy for avoiding the miscounting of aerosols as INP as further discussed below."

Secondly, based on referee comments, section 3.7 has largely been rewritten.

*Referee                                  Comment:*                                  The justification on using a linear regression model and the implicated assumptions on the data
is entirely missing and only legitimated by indicating that other work has used linear regression models. Please add the justification for doing so.

10  *Authors' response:* Linear regressions are commonly used when determining how to use a new technique to determine an atmospheric quantity by relating a measured parameter or parameters to a ground truth measurement.
*Authors' changes in the text:* To explain to the user that there is a wide array of applications for a linear regression, several more papers that use a linear regression for various purposed (Zimmerman et al. 2017;

15  Brunner et al., 2015; Choi et al., 2016) are now cited.

*Referee  Comment*:  Also expand on how the choice of  another  depolarization  ratio  threshold  does influence your results.
*Author response*: Please see the rewritten section 3.7

*Referee        Comment*:        Lastly, the comparison of the TAMU data to the CSU data stays unclear. As presented in the current manuscript the usage of different cut sizes due to instrumental
differences is irritating and needs clarification.
*Authors' response:*

25  The TAMU and CSU CFDCs are independent custom-built instruments which operate on the same principles. Most importantly, because the CSU CFDC doesn't experience WDBT until a higher RH, this comparison provides a means to evaluate performance of the new method under conditions which our traditional method is clearly failing.  In general, because the two CFDCs differ in dimensions, flow rates,

operating conditions (temperature and supersaturation) in the growth and evaporation regions within the instrument, and the choice of detector and size cut-off, an intercomparison is worthwhile.

Most importantly, because the CSU CFDC doesn't experience WDBT until a higher RH, this comparison provides a means to evaluate performance of the new method under conditions which our traditional method is clearly failing. Also, in general, because the two CFDCs are quite different instruments, an intercomparison is worthwhile.

We emphasize that the choice of different size cuts are justifiable, because the two CFDCs are not the same instrument in some key regards. For example, the TAMU CFDC growth chamber is smaller and the residence time is shorter. Therefore ice particles are not expected to grow as large as in the CSU unit. Logically, a small size cut may be more appropriate for the TAMU instrument. However, there are competing parameters, making this a non-straightforward choice, which is why multiple size cuts have been included for consideration in Fig 11. This is an issue that any operator of this type of instrument is concerned with.

*Authors' changes in the manuscript.* For emphasis we include the following statement on pg 24, ln 20, "Thus, inclusion of the CSU data provides a test of the new method at higher relative humidities under conditions when data obtained through the TAMU CFDC's traditional method is clearly spurious due to water droplet breakthrough."

Also, regarding differences in the 2 CFDCs, on pg 9, ln 14-20, the text now reads, "CFDCs in use today are custom-built instruments which vary in physical dimensions and choice of detector, although all operate under the same basic principles. Due to the combination of different chamber dimensions, flow rates, operating conditions (temperature and supersaturation) in the growth and evaporation regions within the instrument, and the choice of detector and size cut-off, WDBT varies from instrument to instrument."

*Referee's comment:*  A quantification of the (range) extension for the operating conditions of the TAMU CFDC when applying the new method along with the associated error should be included.

*Authors' response*:

5 The specific conditions of WDBT vary with CFDC temperature, the ambient humidity, the hygroscopicity of sample aerosols, the size of sample aerosols, and the sample flow which determines the residence time in the instrument.

*Author's changes in text:* This is now stated in the text on page 9, ln 9 as written, "Specific conditions of WDBT vary with CFDC temperature, the ambient humidity, and the hygroscopicity of sample aerosols,

10 the size of sample aerosols, and the sample flow which determines the residence time in the instrument. Typically, in the TAMU CFDC the onset of WDBT occurs at 3 % to 4% $SS_w$, but has been observed as low as 1 % $SS_w$ and as high as 8 % $SS_w$."

**Specific comments:**

15 P. 1-3, Introduction: The authors state the goal of the presented paper to be the development of a new method to quantify INP through a more reliable (phase) discrimination of particles exiting a CFDC, especially when operated under WDBT conditions (cf. p. 1, l. 15-18, p. 4, l. 13-19). In the introduction, the authors carefully describe the importance of ice and mixed-phase clouds and go on to discuss different ice nucleation pathways and INP characteristics (p. 2, l.  3-

20 13 and p. 2, l. 19-20). However, the succinct discussion of ice nucleation mechanism and mixed phase clouds  are integral parts of the discussion on the topic of INP so they are not been removed.

After a brief discussion about the hydrometeor discrimination by LIDAR measurements using depolarization signals (p. 2 l. 25 – p. 3 l. 8) the authors give a detailed

25 overview of the CFDC history and the improvements done to CFDCs (p. 3 l. 1625). None of the   topics mentioned above adds significant information to the topic discussed in the  article,  namely the cor rect discrimination of cloud particle type (phase).       However,       the       introduction       misses a       clear       description       of       the       current       limitations       of       particle       phase

discrimination in CFDC studies as well as a motivation how such limitations affect past and current INP measurements, using CFDCs.

*Authors' response*:

5   The introduction has been significantly revised with the section on particle discrimination moved to early in the introduction.

However, given that this is a study on improvements in ice nucleation instrumentation, we feel the historic details lend important context to the issues, especially those related to water droplet breakthrough and the

10  improvements our new methods contributes.
Hence, we keep the majority of the text and we have added additional details specifically on strategies instruments through history have used to differentiate between ice crystals, water droplets, and aerosols. Also, we would be remiss to leave out (or delete) the section defining ice nucleation mechanisms, so that section remains.

Traditionally phase discrimination has relied on differences in particle size.   An impactor is used to physically eliminate aerosols larger than a certain point (~ 1.75 micrometer diameter).   Traditional detectors are optical particle counters which detector particles in a range of sizes.
*Author's changes in text:* Please see the revised and reorganized introduction.

*Referee*                                                                                               *Comment:*
P. 4 l. 413 give details about how other studies differentiate particle phase, without discussion of the ge neral limitations.  Without this discussion it becomes very hard for the reader to correctly
judge the quality of available  CFDC  data  and  recognize  the  need  for  development  of new

25   instrumentation to improve discrimination of hydrometeor type. I suggest to add some references here a s well.

*Authors' response:* Please see our response to the next comment below. It appears that as the reviewer read further s/he found the answers to his/her questions. To make things clear sooner, we have reordered Sections 1 and 2 of the text, as discussed above.

5 *Referee* *Comment:*

Finally, the benefit of new methods, as described in the presented study becomes clearer.

*Authors' response:* Thank you. We are glad that this section clarified our motivation for new method development, and have moved that section forward in the text.

10 *Referee* *Comment:*

I recommend major changes to the introduction of the presented paper by considerable

shorten or remove some of the topics mentioned above and focusing on background needed

to understand the (size dependent) discrimination of particle phase and associated limitations, to better p

ut the current study into context.

15 *Authors' response:* Please see above in response to specific changes we have made.

*Referee Comment:* P. 1, l. 12: Please change "observed" to "measured".

*Authors' response:* The text is unchanged.

20 *Referee Comment:* P. 1, l. 15: Please change for clarification: "…under which discrimination of hydrometeor phase and thus determination of INP concentrations based on hydrometeor size fails."

*Authors' response:* Okay.

*Authors' changes in the manuscript:* The text now reads," During WDBT, the standard procedure of

25 counted counting all particle which grow beyond the size cut-off as ice crystals fails, which large droplets are miscounted as ice."

*Referee Comment:* P.1, l. 18-19: Please clarify this statement. It is not a challenge

of WDBT that needs to be overcome, as WDBT forms an integral component of any
CFDC study if operated at given conditions, but rather the challenge to reliable
discriminate particle phase of the particles exiting a CFDC once WDBT conditions
are met.

5 *Authors' response:* Okay.

*Authors' changes in the manuscript:* Revised to read, "To accurately measure INP during WDBT..."

*Referee Comment:* P. 1, l. 25: Please change "complicated" to "complex".

*Authors' response:* We prefer the original. The text is unchanged.

*Referee Comment:* P. 1, l. 26: Please clarify whether "precipitation" refers to
spatial/temporal distribution      of     precipitation, precipitation formation or precipitation in general

*Authors' response:* As written, precipitation in general is implied,.. "Because of their complicated
microphysical properties, ice clouds and mixed-phase clouds pose challenges in understanding our global

15 radiative budget and precipitation."

*Referee Comment:* P. 2, l. 2: Leave out "our".

*Authors' response:* Done.

*Referee Comment:* P. 2, l. 8: Leave out "becomes"

20 *Authors' response:* Done.

*Referee Comment:* P. 2, l. 11: Please insert: "aerosol particle…"

*Authors' response:* Done.

*Referee Comment:* P. 2, l. 12: Please change to: "aerosol particle collides with a
supercooled water droplet and …"

*Authors' response:* Since, as the next sentence in the text states, *"*While the exact mechanism of contact freezing remains unresolved, it has been shown that the presence of an INP positioned at a droplet surface facilitates freezing at temperatures several degrees warmer than immersion freezing with identical INPs (Fornea et al., 2009; Durant and Shaw, 2005).*"*, we feel the original is more accurate.

*Referee Comment:* P. 2, l. 20 : Delete "Field"
*Authors' response:* Done.

*Referee Comment:* P. 2 , l.23: Delete "other"
10  *Authors' response:* Done.

*Referee Comment:*  P. 2, l. 28: Delete "can"
*Authors' response:* Done.

15  *Referee Comment:* P. 3, l. 4: Please change to: "…components of the LIDAR
signal retrieved from…"

*Referee Comment:* P. 3, l. 1315: The first argument only applies to field measurements, when
CFDCs are used to characterize ambient INP  concentrations.  However, the data you present
20  here result  from  laboratory measurements, where the number of aerosol particles entering
the cloud chamber (and thus the number of INPs) can be varied by  the experimentalist, making
this argument irrelevant for this study. Please revise this section by making it clearer, that this
is particularly  a limitation of CFDC field studies.
*Authors' response:* We thank the Referee for pointing out that the first sentence here was our place in this
25  paragraph.  It has been moved to the optical section of the introduction.

*Referee Comment:*  P. 3, l. 1022: Shorten this paragraph and to keep the focus on the topic of your
manuscript.

*Authors' response:* As stated above, the introduction is significantly revised and rearranged. This paragraph no longer exists in its original form.

*Referee Comment:* P. 3, l. 23: Please change to "… (CLIMET Inc., Model No. CI3100) …"

5   *Authors' response:* Done.

*Referee Comment:* P. 4, l. 13: Delete "to detect INP"
*Authors' response:* Revised to, "to determine INP concentration."

10   *Referee Comment:* P. 5, l. 4: Please change to: "… are generated, suspended in dry synthetic air…"
*Authors' response:* Done.

*Referee Comment:* P. 5, l. 7: Please specify whether aerosol particles or populations of ice crystals and cloud droplets have been sampled from AIDA.

15   *Authors' response:* Thank you. This is an important distinction.
*Authors' changes in the manuscript:* Revised to read, "During FIN-02, prior to expansion, aerosols were drawn from the AIDA chamber by the various ice nucleation instruments. Following the aerosol sampling period, an AIDA expansion was performed so that INP concentration determined by AIDA could be compared to results from the various visiting instruments."

*Referee Comment:* P. 5, l. 9: Please add: "… of the TAMU CFDC-CASPOL measurements …"
*Authors' response:* Done.

*Referee Comment:* P. 5, l. 25: Please change "limited" to "small".

25   *Authors' response:* Done.

*Referee                                                                                                      Comment:*
P. 6, l. 14: Specify how ice saturation is maintained in the evaporation section of the CFDC

given that you have hydrophobic Teflon walls.    How much are the ice crystals evaporated when passin
g through the lower most 25 cm of the chamber? Can you show that the ice crystals
remain in the sample flow?

*Authors' response:* As cited in the text, experimental details and a full description of the development and
5    characterization of the TAMU CFDC are provided in the references below:
Glen, A., and Brooks, S.D.: Single particle measurements of the optical properties of small ice crystals
and heterogeneous ice nuclei, Aerosol Science and Technology, 48(11), 1123-1132, 2014.
and
Glen, A.: The development of measurement techniques to identify and characterize dusts and ice nuclei
10   in the atmosphere. Diss. Texas A&M University, 2014.

*Referee*                                                                                         *Comment:*

P. 6, l. 4: Please specify why the droplets in some cases only partially evaporate. The
evaporation efficiency is a function of particle residence time in the evaporation section. Your
15   description of TAMU is missing a statement about  the  flows and thus residence times used
 within the TAMU CFDC. Such a discussion is only very briefly given on p. 7, l. 7-8
 and should be          moved to the description of the TAMU operation.         Details of the residence
time are also required to understand how the authors are able to grow ice crystals as large as
40 µm in the CFDC, as suggested by Fig. 6.
20   *Authors' response:*

By definition, when droplets only partially evaporate, the chamber is under WDBT conditions.
Causes on WDBT have been discussed in detail above and earlier in the text.  Ideally, no droplets should
survive the evaporation region of the instrument, but given that WDBT is a problem in this and many
other ice chambers, we see that in practice this is not the case. There are many possible reasons. For
25   instance, droplets may not come to equilibrium prior to existing the chamber under very moist ambient
conditions.

 CFDC flow conditions were already stated in the original text on page 7. "Two mass flow controllers are
used to set the total flow and recirculating sheath flow through the chamber. The difference between the
30   total and sheath flows determines the sample flow. For this campaign, the total flow was set to values

ranging from 6 to 9 L min$^{-1}$ and the sheath flow was set to values ranging from 4 to 7 L min$^{-1}$ resulting in a sample flow that was typically ~2 ± 0.5 L min$^{-1}$."

In Figure 6, direct CFDC measurements are reported. The particles detected (not implied) by the CFDC do include 40 micron diameter particles in size. Ice growth calculations indicate that ice crystals may grow rapidly in size in the chamber (Rogers, 1988; Glen, 2014). Additionally, a known source of large ice crystals are shards that break off the chamber walls occasionally.

*Author changes in manuscript:* Changes referred to here are all parts of the revision discussed in reference to prior comments.

*Referee Comment:* P. 6, l. 8-15: This description of cloud chamber preparation does not add to the topic discussed in the presented paper and should be moved to a supplement.

*Authors' response:* We respectfully disagree. When a manuscript employs an instrument and experimental procedure which are previously published in detail, there is always a delicate balance between re-reported what has been well documents in previous work or not providing enough basic details for a reader to follow the current manuscript. In this case, the referee has asked for additions experimental details above and here asks for fewer details. We do not think it wise to remove the details included in the original.

*Referee Comment:* P. 6, l. 17: Please change to: "… (CLIMET Inc., Model No. CI-3100)…"
Authors' response: Done.

*Referee Comment:* P. 6, l. 26: Please change to: "…backward scatter detector…"
Authors' response: Done. Thanks for pointing this out.

*Referee Comment:* P. 6, l. 5: The position of the mass flow controllers should be specified. I assume these are located downstream of CASPOL?

*Authors' response:* For clarity, the text has been revised. For a schematic, please see Glen and Brooks, 2014a.

*Authors' changes in the text:* The text on page 7 ln 14016 now reads "...the CASPOL is installed at the base of the chamber. Two mass flow controllers downstream of CASPOL are used to set the total flow and recirculating sheath flow through the CFDC-CASPOL. The difference between the total and sheath flows determines the sample flow."

*Referee Comment:* P. 7, l. 13: Please change to: "Temperature, …"
*Authors' response:* Done.

*Referee Comment:* P. 7, l. 17: Please change "ahead" to "upstream"
10  *Authors' response:* Done.

*Referee Comment:* P. 7, l. 119: Please specify how the background (BG) signal from the CFDC is taken into account in more detail. Given that the supersaturation at the position of the aerosol lamina is different before and after a RH scan, the background
15  signal is likely to change from before to after the measurement. The statement between lines 16-19 suggest that there is not always a BG measurement before and after each RH scan ("and/or after"). This makes it hard to follow what BG signal is subtracted from your CFDC-CASPOL measurements.

20  *Authors' response:* Okay, the text is now clarified as below. In our experience, RH doesn't appear to cause a large difference in background signal.
*Author changes in manuscript:* The text now reads "The background period that is closest to a given 1-minute sample period is then applied by subtracting that background concentration from the total concentration measured by the CASPOL at the sample time."

*Referee                        Comment:*                        P. 7, l. 20-
24: This sentence is misleading. I assume you refer to the usage of the optical particle counter and the associated size cutoff used to discriminate between ice crystals and

cloud droplets when using the term "traditional analysis". This is in contrast to the p. 6, l.
16- 19, where it is described that TAMU had been used with both, OPC and CASPOL, so in
principle both detectors can be interpreted as the traditional detector technique/analysis
method. I suggest to make a clearer distinction between these two cases (OPC vs. CASPOL as
detector) and give a clear statement earlier in the manuscript what the "traditional analysis" refers to. *Au
thors' response:* Yes, thanks. We see how this could have been confusing in the original text. Actually,
we refer to any size-discrimination (by OPC or CASPOL forward scatter detector) as traditional analysis,
and the use of the depolarization as a new method.

*Author changes in manuscript:*

p. 4 ln 12: "Particles are sized according to the intensity of light which reaches the CASPOL's forward
scatter detector, as in a traditional OPC."

and

p. 8 ln 6: "During FIN-02, data collected by the CASPOL's forward scattering detector was used for the
traditional analysis."

*Referee*                  *Comment:*

P. 7, l. 27: This statement is misleading. There are no limitations of the OPC technique (discrimin
ation purely based on size) discussed in Section 2.3. Please delete the part in
brackets. The authors start a superficial discussion of the limitations by using and OPC and a
size threshold to discriminate the phase of cloud hydrometeors at various points of their manus
cript, e.g. p. 3. l. 15-16, p. 3, l. 23-25. However, a clear statement that under certain
Thermodynamic conditions within the TAMU CFDC, cloud droplets and ice crystals of the same size ca
n be present, thus biasing a pure phase discrimination based on particle size, is missing.
This should be discussed in the introduction.

*Authors' response:* For a detailed discussion of the many causes of WDBT and related instrument details,
please see page 8 ln 14-22, which have been expanded and revised. Note, however, that we do not refer

to thermodynamic conditions, however, because WDBT is consistent with failure to remove large supercooled drops which do not reach thermodynamic conditions by the time they reach the chamber.

*Referee* *Comment:*

5   P. 8, l. 3: The authors mention the limitations of traditional methods, but do not discuss differences how the ice crystal size threshold may be chosen. Please give more details.

*Authors' response:* Ice crystal size thresholds have been chosen empirically based on laboratory results. For further details please see our previous work (Glen, 2014B).

10  *Referee Comment:* P. 8, l. 8: What do the authors mean by positive or negative artifacts?

*Authors' response:* This was a mistaken choice of words, as noted by 2 referees. In reality only positive artifacts are possible. "Positive artifacts" mean water droplets breaking through are counted as members of the ice crystal population. "Negative artifacts" would mean ice particles not counted because one thinks they are water droplets, but in practice there is no way for that to occur.

15  *Authors' changes in manuscript:* "if the instrument is unintentionally operated at supersaturations above WDBT, droplets will be miscounted as ice crystals."

*Referee* *Comment:*

P. 8, l. 12: The challenges are not really presented by WDBT, but are rather inherent to any

20  optical method that uses size as a means of phase discrimination.

*Authors' response:* This is the second time the referee mentions this issue, which is really a word choice issue. We revised the language in the abstract as per his/her recommendation. For clarity, we feel it best to keep the original text here, as this manuscript is addressing a measurement challenge specifically occurring when WDBT occurs.

*Referee Comment:* P. 8, l. 14-21: This section is partly a repetition of the statement made on p. 5, l. 23. Besides, it should be clear to any reader that a particle of any type (aerosol, cloud droplet) larger than the cut size will be misclassified as ice crystal by the OPC

when using size thresholds to define an ice phase.

*Authors' response:* Given the importance of this issue, we keep this section on page 8, but have shortened it.

*Authors' changes in manuscript:* Pg 8 now reads, "Although operation with an upstream impactor reduced this problem, ~1 to 10% of particles larger than 2 μm (depending on flow) may make it into the chamber to contribute to the apparent INP signal."

*Referee          Comment:*          P. 8, l. 19-21: Are the authors trying to say that large aerosols are not counted as ice crystals  in  their detector and they can be distinguished from an ice crystal of the same size?

*Authors' response:* At this stage in the manuscript, the text only states that such capabilities would be an improvement, given the limitations of the traditional analysis.

*Authors' changes in manuscript:* The text now reads, "A new analysis method that differentiates between large aerosols and ice crystals is needed since it would remove the need to limit the size of particles allowed into the instrument in the first place."

*Referee                                                                                      Comment:*

P. 8, l. 16: The expression "higher supercooled temperatures" is not clear. The authors should indicate more clearly what they compare to and point out reasons why the new analysis method is particularly powerful at higher T. The only indirect hint for this is given by the statement in brackets on p. 8, l. 10.

*Authors' response:* The statement in question was deleted in response to the Referee's previous comment.

*Referee Comment:* I recommend moving the discussion of section 2.4 to the Introduction to motivate the development of the new method.

*Authors' response:* Please see above that we have significantly revised the introduction. Moving this particular section was something the authors previously discussed. We decided that so much detail about challenges specific to our CFDC would be better left in the experimental section.

*Referee*                                                             *Comment:*

P. 8, l. 25: Please be more general in a first statement. The goal, as far as I understand from the present

ed study, is to first distinguish more accurately between aerosol particles, ice crystals and cloud

5   droplets and then in a second step quantify the INP, as you clearly write e.g. p. 4, l. 13.

*Authors' response:* We see the referee's parts 1 and 2 as parts of the same objective. We prefer to keep

the original text here.

*Referee*                                                             *Comment:*

10  P. 9, l. 1: I suggest repeating the meaning of the different parameters again. E.g. "Similar to eq.

(1) $B_{\perp,CAS}$ and $B_{\parallel,CAS}$ denote the perpendicular and parallel components of the backscattering signal

, respectively, and the subscript CAS refers to the CASPOL signal…."

*Authors' response:* This is a good suggestion.

*Authors' changes in manuscript:* Done.

*Referee*                                                             *Comment:*

P. 8, l. 24: Section 2.5 describes CASPOL instrumental details and should be moved to

the description of CASPOL in section 2.2.

*Authors' response:* This is a good suggestion. Due to other suggestions the CASPOL description is now

20  in Section 2.2. and Section 2.5 has been moved to that section.

*Referee Comment:* P. 9, l. 21: Please explain why the neutralizer prevents particle loss.

*Authors' response:* Charged particles are attracted to the walls of the tubing.

*Authors' changes in manuscript:* This statement is now including in the text..."to prevent particle loss

25  since charged particles tend to be attracted to the walls of sample tubing."

*Referee Comment:* P. 9, l. 22: Please change to: "the" before CASPOL

*Authors' response:* No. The grammar is correct in the original.

*Referee Comment:* P. 10, l. 15: I assume you are referring to CFDC-CASPOL measurements,

*Authors' response:* Yes.

*Authors' changes in manuscript:* Changed CASPOL to CFDC-CASPOL.

*Referee Comment:* P. 10, l. 20: Please clarify the source of your temperature uncertainty here. How can the temperature uncertainty here be much lower than the value given on p. 7, l. 10?

*Authors' response:* This is reported instrument uncertainty, whereas on 7, we reported the range of temperatures over which collected experimental data was included in the intercomparisons. Specifically, temperature here is based on experimental temperature, derived from a set of 8 thermocouples calibrated to a reference RTD.

*Authors' changes in manuscript:* The text on pg 7 has now been clarified to explain that the temperature range on pg 7 was not a report of instrument uncertainty. Instead, it was the range of operating temperatures of measurements included in the FIN-02 intercomparison.

*Referee*                                                  *Comment:*

P. 10, l. 15: Please change SS to saturation ratio formulation throughout the manuscript. This will avoid confusion as of the negative sign and make your figures more easily readable.

20 *Authors' response:* We feel that supersaturation is useful because 0 % demarcates when water droplets may begin to form in the chamber. Also, SS is often used in ice nucleation papers.

*Authors' changes in manuscript:* None.

*Referee*                                                  *Comment:*

25 P. 10, l. 20: Do you suggest that particles smaller than 2 µm are not necessarily frozen?

*Authors' response: 2* µm is the nominal size cut for ice. Both calculations and experimental tests have shown that if size nucleation in our chamber, ice will grow to above 2 µm (Glen, 2014.)

*Referee Comment:* P. 10, l. 23: Insert comma after "datasets"

*Authors' response:* Done.

*Referee Comment:* P. 11, l. 4- 5: Please clarify the usage of optical signatures by
Hu et al. (2009) and how this relates to your study.

5   *Author's response:* Revised for clarity. Hu et al is a successfully example of using backscatter and
depolarization data to determine cloud particle phase.

*Authors' changes in the manuscript:* "In an analogous method, optical signatures produced from
CALIPSO satellite backscatter and depolarization data have been used to identify cloud phase (Hu et al.;
2009)."

*Referee Comment:* P. 11, l. 6: Delete "training"

*Author's response:* Training has a very specific meaning here, so we choose to keep it.

*Referee Comment:* P. 11, l. 13: Please clarify whether Dp refers to the optical diameter

15   measured with CASPOL or to another diameter measured with another device.

*Author's response:* For clarity Dp has been replaced with diameter and revised the text as below:

*Authors' changes in manuscript:* "As discussed, the ice crystal and droplet training data shown in Fig. 1
only includes particles with optical diameters $\geq 2$ μm and $\geq 1$ μm, respectively.

20   *Referee                                                                 Comment:*
P. 11, l. 17: Please extend your interpretation of why almost only ice crystals show high values for B☐/
F and what that implies.

*Author's response:* Here we report a direct observation from data in the figure.

*Authors' changes in manuscript:* No change has been made.

*Referee Comment:* P. 11, l. 23: Insert point after "et al."

*Author's response:* Done.

*Referee*                                                                                                          *Comment:*

P. 11, l. 24 : Please clarify why this is an empirical tool and how this affects the application to

your data.

*Author's response:* The optical signatures are used to detect patterns in backscattering vs. depolarization

5   plots for different particle types. By definition, these observed differences (if found) are empirical rather

than theoretical.

*Authors' changes in manuscript:* No change has been made.

*Referee*                                                                                                          *Comment:*

10   P. 12, l. 4: "It is assumed that the CASPOL emits an incident beam that propagates along the  z…"  Wh

y is it only assumed? Can you verify this experimentally?

*Author's response:*  The Referee has a good point. This is a reality, not an assumption.

*Authors' changes in manuscript:* Deleted, "It is assumed that."

15    *Referee Comment:* Which direction is the z direction? A schematic figure defining the different parts of

CASPOL along with a coordinate system will definitely improve your description here.  Please

 add a figure to your supplement.

*Author's response:*  As clearly stated in the text- z is the direction on propagation of the incident CASPOL

laser beam. See page 12, ln 24, "The CASPOL emits an incident beam that propagates along the z

20   direction in the form."  Also, schematics of the CASPOL have been previously published in Glen and

Brooks 2013 & 2014.)

*Authors' changes in manuscript:* No change has been made.

*Referee Comment:* It is not the CASPOL, but the laser diode of the CASPOL that emits the light.

25   *Author's response:*  True, "laser" now added.

*Referee Comment:* P. 12, l. 8: Please change to: "…line linking particle (position) and detection point."

*Author's response:* We feel that this would not be an improvement.

*Referee Comment:* P. 12, l. 12: Please insert commas: "… ratio, δModel, can …"
*Author's response:* Done.

5 *Referee Comment:* P. 12, l. 15-16: Please insert commas: "…matrix, Pij, the amplitude matrix, Sij, and the scattering  cross section, Csca, …"
*Author's response:* Done.

*Referee Comment:* P. 13, l. 13: Please replace "vs." by "as a function of"
10 *Author's response:* Done.

*Referee Comment:* P. 13, l. 17: Please delete "the" in front of optical signatures.
*Author's response:* There is no "the" in the line specified.

15 *Referee Comment:*  P. 13, l. 13-21: Please elaborate this discussion and give more details:

*Author's response:*  This section has been  rewritten and expanded. Also, the range of particle and ice diameters have been expanded in Figure 2,as discussed in the new text.
*Authors' changes in manuscript.* Please see the revised section on Pg 14 ln 15 to pg 15, ln 5, and Figure
20 2.

*Referee* *Comment:*
Below approx.  2 µm  no modeled  depolarization  ratios are given for any  of  the ice  crystals, making a comparison between aerosol particles and ice crystals as suggested  in the text
25 difficult (l. 18-19)
*Author's response:*  This is an excellent point. Calculations for a wider range of ice crystal sizes are now included and discussed. See previous response.

*Referee Comment:*

The authors discuss the differences in depolarization ratio as a function of ice crystal habit in the range 2- 4 µm. However, there is a clear distinction also above 10 µm for e.g. hexagonal plates and hexagonal columns. This needs to be explained.

5 *Author's response:* Please see our response above. This section has been expanded. The differences at larger diameters are mentioned in the text, although there is not a theoretical explanation for the observed differences.

*Referee Comment:*

10 What are the uncertainties associated with the modeled results. Errors bars should be included for the individual data points to render a comparison possible at all.

*Author's response:* Errors bars are not available. This is a tricky question. In the case of the modeling, the model is highly accurate for the chosen inputs. The uncertainty arises from assignments of the correct

15 inputs. In this case, by far the largest uncertainty in the modeled results in the choice of shape. This is way we include 3 shapes. Other inputs, including wavelength and particle diameter, refractive index are known with high precision.

*Authors' changes in the manuscript:* The uncertainty which arises due to particle shape is now explicitly states, "It is not known which of these habits best represents individual ice crystals nucleated and grown

20 in the CFDC. Fortunately, if it is assumed that only particles of 2 µm diameter or larger are ice crystals in the CFDC, these theoretical results shown that all water and ice particles on any of the three habits will be accurately identified."

*Referee Comment:* P. 14, l. 1: Please insert: "aerosol particles"

25 *Author's response:* Since "aerosols" is acceptable grammar, we prefer to keep it. This is unchanged.

*Referee Comment:* P. 14, l. 2: Please insert: "… shown in Fig. 1 …"
*Author's response:* Done.

*Referee*                                                                                   *Comment:*

P. 14, l. 4: Please change to: "Each nominal droplet size produced by the VOAG is treated as a

separate population in the training data set and …"

5   *Author's response:* We consider the original text to be more succinct in this case.

*Referee Comment:* P. 14, l. 5: Please change to: "… in Fig. 1a …"

*Author's response:* Done.

10   *Referee*                                                                                 *Comment:*

P. 14, l. 10: Please clarify how the selection criterion for ice crystals (depolarization ratio > 0.3)  is deriv

ed and how it is connected to the values discussed in Fig. 1 (cf. p. 11, l. 15).

*Author's response:* It can be seen qualitative in the figure that droplets have depolarization ratios up to

0.3. At this point in the manuscript, this is only a simple choice based on visual observation. However, in

15   Section 3.7, optimization of the depolarization threshold is performed using linear regression analysis,

and the results come to the same conclusion, that 0.3 is the preferred choice of nominal threshold.

*Author's changes in manuscript:* We have added text (pg 15 ln 20), that states "It can be seen from this

figure that droplets have depolarization ratios up to 0.3. Therefore, we visually assign 0.3 as the nominal

depolarization threshold cut-off for differentiating between ice crystals and non-ice particles.  The choice

20   on 0.3 is further evaluated in Section 3.7 below."

*Referee*                                          *Comment:*                                          P. 14, l. 15-

16: Why do the aerosol particles in Fig. 1c show a mode only in the constrained size

range between 5 to 10 µm and not above 5 µm in general?

25   *Author's response:* Fig. 1c is not discussed at this point in the manuscript, so we are unsure of the

Referee's intended question.

*Referee* *Comment:*

P. 15, l. 12: Please specify what you mean by "the size mode". I think you are referring to the smaller mode of the bimodal size distribution described above.

*Author's response:* Yes, that's correct.

5 *Authors' changes in manuscript:* Changed "the size mode" to "aerosol size distribution."

*Referee Comment:* P. 15, l. 13- 19: In section 3.3 you discuss the usage of a depolarization Threshold of 0.3 to discriminate between different particles types ("nominal selection criteria for depolarizing ice crystals"). In Fig. 4b all of your particles have significantly lower

10 depolarization values, even at times when you are supersaturated. Please clearly state, that water droplets cannot be present during the time period before 11:55 due to the fact of being sub saturated, to avoid any confusion with your threshold of 0.3 discussed earlier.

*Author's response:* Actually, the value here is the mean depolarization reported and it is consistent with the mean depolarization of training data ice crystals. As indicated on the y axis label, Figure 4b shows

15 the mean depolarization ratio of all particles above 2 microns at that time. Because we only consider those particles larger than 2 microns and we are not in WDBT conditions until 11:55, these particles are ice crystals.

*Authors' changes in manuscript*: We have expanded the previous section that discusses mean depolarization ratio of the training datasets to reduce confusion.

*Referee* *Comment:* P. 15, l. 16-

18: This is not correct. It is not the mean depolarization ratio, which has a strong dependence on whether WDBT is occurring in the CFDC, or not. Analyzing the depolarization ratio, you can observe the moment when WDBT occurs in the CFDC. Please phrase that more

25 carefully.

*Author's response:* See previous comment.

*Referee*                                                    *Comment:*

P. 16, l. 17: The statement "… at colder temperatures of these runs" is misleading, as the center
 temperature in your CFDC stays constant for each of the two runs. Further, the two Snomax®
Cases presented are not labeled in the figure, such that the reader cannot assign a CFDC c
5   enter temperature difference between the runs from the lines in Fig. 5.

*Author's response:* We agree this should be clearer.

*Authors' changes in manuscript:*   The text is now modified.   In addition, we have added labeled,
"Snowmax 21 ºC, and Snowmax 33 ºC to figure 5.

10  *Referee*                          *Comment:*                          P. 16, l. 26-

27: How does the error shown for the observed values compare to the instrumental
uncertainty from CASPOL to determine the right depolarization ratio? Please add
error bars associated  with  the  modeled  results.  Consider using standard  error  of  the
 mean for  normalization to number of observed particles at the different sizes.

15  *Author's response:*  Please see our responses above regarding the challenges of reporting error for the
modeling results.

 *Referee*                                                    *Comment:*

P. 16, l. 28: Please  add  for  clarification: "…  from  all  FIN02  experiments  and  not  only
20  the Snomax® experiments discussed in Section 3.5."

*Author's response:* Done.

*Referee Comment:* P. 17, l. 1: Remove "u" after 2.

*Author's response:*  Done.

 *Referee*                          *Comment:*                          P. 17, l. 1-

4: Consider deletion as you already reference to the description given in section 3.5.

*Author's response:* Although this is somewhat redundant, we feel it best to restate these rules to avoid any confusion.

*Referee Comment:* P. 17, l. 510: Do the authors have any idea what type of ice crystal

5 is formed in the CFDC? Is this dependent on the aerosol type/experiment?

Which of the modeled ice crystals is closest to the "CFDC ice crystals"?

*Author's response:* Unfortunately, no. We see a wide variety in backscatter and depolarization ratios and don't have any way to answer assess this. As an aside, we have tried collecting ice crystals exiting the CFDC in plastic casts (made from dissolving plastic in dichloroethane), but our attempts so far data were

10 not of high enough quality to determine ice crystal habit.

*Referee Comment:* P. 17, l. 14: Please replace "region" by "population".

*Author's response:* Done.

15 *Referee Comment:* P. 17, l. 19: Why do the "CFDC ice crystals" show depolarization ratios $< 0.35$ for all sizes shown? It is unclear to me how this relates to the data shown in Fig. 5, where the majority other ice crystals show larger depolarization ratios. In addition, none of your "CFDC ice crystals" would meet the 0.3 threshold in depolarization ratio discussed on p. 14. Please explain. Is this due to averaging over all FIN-02 experiments?

20 *Author's response:* Please look again at the figure 5 and the related discussion. The manuscript states the opposite of what the referee has said. As stated "13.5 % of ice crystals in the CFDC achieve a depolarization ratio $> 0.3$, compared to 1.5 % percent of water droplets and 0.3 % of aerosols. Additionally, please note the figure 6 is showing mean values depolarization ratio. Since many of the particles detected have relatively low depolarization ratios (see figure 5 and figure 3b), this value will be

25 low. We've added the mean throughout the discussion of figure 6 to clarify this.

*Author's changes in the manuscript:* "In this section, modeled and observed particles discussed in the preceding results section are compared. Fig. 6 shows modeled and observed mean depolarization ratios

of particles…" "In Fig. 6, both the model calculations and the observed results indicate that ice crystals have higher mean depolarization ratios…"

*Referee Comment:* P. 17, l. 23-26: More details about the "underestimation of the depolarization by
5  CASPOL and the detection limit" along with an appropriate reference should be given. Should the under estimation of depolarization by CASPOL not preliminary affect smaller sized particles (that scatter in relatively less light)? Thus the discrepancies between modeled and observed results should decrease as a function of size, as the detection by CASPOL becomes more reliable?

*Author's response:*  In general, particles scatter relatively little perpendicularly polarized light in the
10  backward 1 raw count which translates roughly a scattering cross section of $\sim 1 \times 10^{-13}$ cm$^2$.  This limit results in the CASPOL registering a perpendicular signal below the CASPOL's detection limit for 45 % of training ice crystals, 76 % of training aerosols, and 57 % of training droplets. In the training data sets, all particles with undetected perpendicularly polarized detector were assigned depolarization ratio of zero.

*Author's change in the manuscript:* The full explanation above, "In general, particles scatter relatively
15  little perpendicularly polarized light in the backward direction, …" is now added to the text on pg 20 ln 12.

*Referee Comment:* P. 18, l. 22-24: How does this statement fit to your data shown in Fig. 6 (cf. "CFDC ice crystals")?
20  *Authors' response:*  This statement cannot be directly applied to figure 6 since the figure shows a mean and error bars that report the standard deviation. Since only 13.5 % of ice crystals achieve a depolarization ratio of 0.3 or greater, the error bars here will not show this range of particles. Since we have focused heavily on this point in the depolarization ratio distributions previously in the manuscript, we do not wish to expand anymore here. Rather, the point of this figure is to compare the mean observed depolarization
25  ratios to modeled depolarization ratios. No change has been made.

*Referee Comment:* P. 18, l. 25: This is contradictory to the values you state on p. 16, l. 8-9. Please clarify.
*Authors' response:*  The referee is right. This was a typo in the original statement here.

Author's change in the manuscript: The text (pg 21, ln 1) now reads, "A depolarization ratio threshold of 0.3 is a favorable criterion to detect ice crystals because < 2% water droplets and aerosols achieve this depolarization ratio."

5 *Referee Comment:* P. 18, l. 27: Please clarify what signal to noise ratio you refer to.

*Authors' response:* The signal is ice crystals, the noise is water droplets with a depolarization ratio of 0.3 or greater.

*Author's change in the manuscript: Pg 21, ln 5:* "…effectively reducing the signal (ice crystals) to noise (water droplets with $\delta \geq 0.3$) ratio ~1:1 or worse."

*Referee Comment:* P. 19, l. 7: I suggest giving more details here, as referring to an "optimal threshold" at this point is confusing. This threshold comes out of your training data sets (Fig. 3). However, in Fig. 6 you show that application of this threshold is not sufficient to discriminate droplets and ice crystals for WDBT conditions anymore. There, using the term "optimal threshold" should be avoided.

15 *Authors' response:* Agreed.

*Author's change in the manuscript:* The text has been modified and the depolarization ratio threshold is not referred to as optimal until after the linear regression fit has been introduced on pg. 22 ln 20. "Figure 7 shows that the 0.35 threshold out performs all other thresholds when $M > 20$. The mean $R^2$ value for the 0.35 threshold is 0.46. The next best performing threshold is 0.3 with a mean $R^2$ value of 0.44.

20 However, aerosol and water droplet concentrations in CFDC experiments are typically in the range $1 < M < 20$ so it is appropriate to give more weight to the performance of the fit at these values. The mean $R^2$ value in this range of $M$ for the 0.3 and 0.35 thresholds 0.71 and 0.7 respectively. While the performance of these thresholds perform comparably over this range, we selected the 0.3 threshold because it will slightly outperform the 0.35 threshold, especially when detecting lower INP concentrations."

P. 19, l. 24-28: Please specify why the linear fit was done for the case of M = 1. It is not clear, why the fit derived from the M = 1 case, is applied to all the other data sets M = 2 to M = 50.

*Authors' response:* The concentration of aerosols and droplets can change in the CFDC. This purpose of this exercise is to understand how that fit will perform over all ranges of *M*. This is stated in the manuscript where we say, "Only one fit is determined for each threshold because we cannot feasibly design a model that adapts to water droplet and aerosol concentration in the CFDC."

5 *Author's changes in the manuscript:* pg 21, ln 26: 1) "The upper range of *M* values here represents an extreme sampling condition where there are many aerosols and many CCN that will form cloud droplets, but not many INP that will form ice crystals. Given the relatively high number of aerosols and droplets, this would represent the most challenging sampling scenario for proposed new method."

10 *Referee Comment:* P. 20, l. 4: Please replace "The Fig." by "It".
*Authors' response:* This sentence was removed during revision of the section.

*Referee Comment:* P. 20 l. 9: Given that you describe an optimization problem, there should be one optimum and a range of acceptable values. Please justify your statement on p. 9, l.2
15 *Authors' response:* This is a good point. Please see our response above. We have added additional values and discussion.

*Referee Comment:* P. 20, l. 5-8: You describe a threshold used to distinguish between ice crystals, droplets and aerosols, to then derive Eq. (9), which yields the number of ice nucleating particles. However, the
20 number of ice crystals is usually way larger than the number of INP. Please explain in more detail, how you derive a "parameterization" for INP at this stage.
*Authors' response:* In the CFDC, we assume a one to one relationship between ice crystals and INP. There is no shattering or multiplication, so this is an accurate assumption. We believe the reviewer is alluding to field observations of ice crystals which have been larger than concurrent INP concentrations.
25 It is far beyond the scope of this manuscript to deal with disagreements between instruments in the literature, and most importantly, that question is not applicable to the internal chamber of the CFDC.

*Referee Comment:* P. 20, l. 22: Please add: "Each relative humidity scan..."

*Authors' response:* Done.

*Referee Comment:* P. 20, l. 23: Please replace: "Supersaturation" by "Saturation ratio"

*Authors' response:* Please see above. We have chosen to keep "supersaturation" as the metric of interest
5  throughout this manuscript.

*Referee Comment:* P. 20, l. 25: Before this statement the meaning of the circles and the asterisks needs
to be introduced in the text.

*Authors' response:* Done.

10 *Author's change in the manuscript:* In section 3.8, we've modified to, "The reported concentrations reveal
that the traditional (circles) and depolarization ratio (*) methods generally agree during "ice only" periods
(blue symbols in Fig. 8)."

*Referee Comment:* P. 21, l. 11: Please specify what the value of the CASPOL uncertainty refers to. Is
15 this the depolarization ratio signal?

*Authors' response:* This is the CFDC-CASPOL uncertainty in INP concentration, based on combined
instrumental uncertainties.

*Authors' changes in the manuscript:* This statement is now included.

20 *Referee Comment:* P. 21, l. 23: Please quantify the detection limit of CASPOL or give an appropriate
reference.

*Authors' response:* As discussed in the experimental section in considerable detail, the CASPOL is a
single particle 60 Hz instrument. Please see Glen and Brooks 2013 and 2014 for characterization of
instrument performance.

*Referee Comment:* P. 21, l. 22: Please replace "polluting" to "biasing"

*Authors' response:* Revised to " large water droplets being miscounted as INP"

*Referee Comment:* P. 22, l. 2: Please add: "...mean percent error (MPE)..."

*Authors' response:* Done.

*Referee Comment:* P. 22, l. 11-12: Please quantify the concentration rate, where the new method is applicable rather than stating "high concentrations" and quantify the "accuracy" indicated.

*Authors' response:* Since accuracy as a function of concentration was just discussed in detail in the previous paragraph of the manuscript, we do not wish to repeat those details here.

*Referee Comment:* P. 22, l. 16: The benefit from this last paragraph and the additional comparison to the CSU CFDC, along with different cut-sizes shown in Fig. 11, does not become clear. Please explain in more detail.

*Authors' response:* Please see the experimental section in which a detailed description of the differences between the two CFDC are discussed. Most importantly, because the CSU CFDC doesn't experience WDBT until a higher RH, this comparison provides a means to evaluate performance of the new method under conditions which our traditional method is clearly failing.

Also, in general, because the two CFDCs are quite different instruments, an intercomparison is worthwhile.

*Authors' changes in the manuscript.* For emphasis we include the following statement on page 24, ln 20 Thus, inclusion of the CSU data provides a test of the new method at higher relative humidities under conditions when data obtained through the TAMU CFDC's traditional method is clearly spurious due to water droplet breakthrough."

*Referee Comment:* P. 22, l. 16 – P. 23, l. 14: Is your new method not applicable to other CFDCs operated along with CASPOL at all?

*Authors' response:* In theory the method is application. No one has tried that yet, to the best of our knowledge.

As the comparison to the modeled results indicates, having the CASPOL's unique particle-by-particle measurements of both depolarization and size is a clear advantage for reliable particle discrimination. Depolarization alone can be used to differentiate between droplets and ice crystals. However, to differentiate between dust aerosol and ice crystals are both depolarizing, so the size information provided
5   by the forward scattering detector is needed as well as depolarization.

Figures:

*Referee Comment:* Figure 1: Please locate the axis ticks also outside of the subpanel boxes to increase readability. Please be consistent with the terminology defined in Eq. (2) and include the subscript "CAS"
10  in the axis labels (also on the y-axis). "CAS" subscript should be included in terminology used in figure caption.

*Authors' response:* Done.

*Referee Comment:* Subpanels (a/d), (b/e) and (c/f) are plotted for the same datasets. However, the color
15  bars for the upper row of subpanels and the lower row of subpanels use different colorcoding, which renders a comparison difficult. I suggest to change this using the same range for the color scale.

*Authors' response:* The color scales for the plots have been carefully selected for readability of the plots. The objective of the plot is to reports patterns in the optical signatures and not to compare them, so it's appropriate that the scales are different in this case. No change has been made.

*Referee Comment:* Figure 2: X-axis (labels) should be read as log-scale.

 Please include model calculations for larger aerosol sizes, such that there is a size overlap for the different particle types. This is needed to justify your statement on p. 13, l. 18.

 Please delete the term "Model" in your legend, as this is redundant information from the y-
25  axis label and the figure caption. Caption: Please insert comma after droplets.

*Authors' response:* The x-axis is already plotted as a log scale. Larger aerosols and smaller ice crystals have now been incorporated into the figure, and "model" has been removed from the legend. The comma has been added to the caption.

*Referee Comment:* Figure 3: Please add symbol for depolarization ratio in x-axis label of panel (b), for consistency. These are all size distributions measured with CASPOL, right? Was there any additional instrument used, e.g. an Aerodynamic Particle Sizer, to verify the size of the produced particles? If so,

5  please add these information and graphs to a supplement.

*Authors' response:* No other instrument was used to size particles here.

*Authors' changes in the manuscript:* To figure 3, we have added "as detected by CASPOL" in the caption, and added d.r. symbol to x-axis of (b).

10  *Referee Comment:* Figure 4:

What are these large particles prior to 10:45 CET? The authors mention (p. 5, l. 24) that no impactor was used during the FIN-02 campaign and that the number of large particles was limited. I suggest to add a number size distribution of the Snomax® sample shown in Fig. 4 to the appendix for clarification. How do these large particles in the range 5- 10 μm influence the depolarization

15  ratio shown in Fig. 4b (see also your Fig. 3)? Please add a description to the discussion in the manuscript.

*Authors' response:*  The large particles here are ice crystals. We've added a size distribution to the supplemental section that shows that there are no Snomax aerosols that are larger than 2-micron diameter.

*Author's changes to the manuscript:* The text now includes the supplement figure, Fig S1. and a statement referring to it (pg 17, ln 6.)

*Referee Comment:*  I suggest showing Panels (a) and (b) as a function of saturation ratio w.r.t. water instead of time. Saturation ratio w.r.t. ice can then be given as a second/top x-axis for instance. There is no additional information given by time. By using saturation ratio w.r.t. water it will be easier for the reader to put the discussed WDBT into context. Indicating ice saturation ratio will help to identify the

25  formation of ice crystals.

*Authors' response:*  This is a good advice. However, there are several challenges presented by the data that inhibit us from displaying the data in this manner. Because the data is not collected at regular intervals

of super saturation, there would be breaks in the data that make the plots hard to decipher and likely confusing to the reader. After attempting to plot the data this way, we decided that it would be better to display the data as we have here.

5   *Referee Comment:*   The text on p. 15 should be changed accordingly and can make more clear what is meant with "normal operating conditions" (p. 15, l.8). Further, labels for "normal" and "WDBT" conditions in Fig. 4 could help.
*Authors' response:*   In the original version of figure 4, there was already a label to describe when WDBT happens.

10   Authors' response: We have added a label for "Normal Operating Conditions" to Fig. 4.

*Referee Comment:* Please make axis ticks more visible (e.g. reduce thickness of axes) and add ticks to x-axes in Fig 4a/b.
*Authors' response:*   Done.

*Referee Comment:* Add explanation for the horizontal dashed lines in the figure caption (see p. 7, l.20).
*Authors' response:*   Done.

*Referee Comment:*   Caption for panel c should include the case number and a reference to Table 1.
20   *Authors' response:*   Done.

*Referee Comment:* Figure 6: I suggest using log-scale for the x-axis.
 Even though you state that the error bars show standard deviations from the mean, they seem
to be on the same order of magnitude. Please add error bars (e.g. for some of the data)
25   *Authors' response:*   The author's are confused about what this comment is requesting since all standard deviations are reported. Additionally, we have confirmed that the standard deviations reported are correct. The x-axis is already reported as a log-scale.

*Referee Comment* Please change the label of the y-axis as the data is a mixture of modeled and observed depolarization ratios.

*Authors' response:* Done.

5   *Referee Comment:* Figure 10:
    The x-axis label should read "traditional concentration".
    *Authors' response:* Done.

**Anonymous Referee #4

This manuscript (Using depolarization to quantify ice nucleating particle concentrations: a new method by Zenker et al.) capitalizes on the ability of the CASPOL detection method to capture the depolarization information from particles, droplets and ice particles in the TAMU CFDC and identify them under different operating conditions. The method may be applicable to other systems but each CFDC is unique.

15   The manuscript includes the development of a new empirical analysis method, to quantify ice nucleating particle concentrations and presents a way to deal with especially the data obtained during water droplet breakthrough, which is difficult to interpret. I believe that the manuscript topic does fall into the scope of AMT. Generally, the paper is readable, the analysis is carefully done and discussion points seem to be well supported by data. There is a limitation for this method in that higher concentrations only obtainable

20   in a laboratory are applicable; the authors are upfront about this limitation.

*Referee Comment:* There are some major considerations that, if addressed, could strengthen the paper: The authors may want to consider strengthening the end of their introduction to describe in more details the trajectory of work presented in the paper. Such a road map is limited here and more details could be

25   helpful. In the body, there is little text regarding the comparison but there is a lot of text with many details regarding the development of the empirical analysis, yet these seem equally weighted in the introduction.

*Authors' response:* As per this Referee's comment as well as those of another, the introduction has been significantly restructured, including a "road map" in the final paragraph of the section, as suggested here.

*Authors' changes in the manuscript.*  Please see the new introduction as a whole.

*Referee Comment:*  In terms of the training data, the text notes that no droplets below 2 µm were studied and this is reflected in figure 3. However, figure 6 shows training droplets at 0.7 µm. This is confusing. Further, since this size is a cut-off point for the analysis, it might be helpful to include smaller particles generation or to explain how the data in figure 6 was observed.

*Authors' response:* I would ask the referee to revisit the figure. No training droplets below 1 micron are plotted. The training droplets are cut at 1 micron to eliminate residual 2-propanol droplets that form in the generation of the olive oil droplets.

*Authors' changes in the manuscript:* To avoid confusion, Figure 6 has now been revised and does not show training droplets below 1 micron. The elimination of  <1 micron diameter residual 2-propanol droplets is also stated in the text.

*Referee Comment:*  There may be minor scientific issues associated with the depolarization theory (that section of the paper was difficult to follow and there seemed to me to be some confusion or missing information associated with representations of matrices, matrix elements and values and/or units). In particular, the section on page 12 surrounding equations 6-7 is especially confusing. The authors note that these equations deal with the amplitude matrix, but then their inclusion in the equation appears to be an element with only one index. Further, it would be helpful to explain this part of the model further. What do these relationships (eqn 6-7) represent? I see how they combine to create eqn 8 but why?

*Authors' response:* The text has been revised to indicate that not only one index is included. Also, equation 8 is required in the form presented here for direct comparison to the CASPOL which detects light over single band of back scattering angles. 168º to 176º.  This was mentioned in the experimental section, but we now include it here as well.

*Authors' changes in the manuscript:* The text on page 13 now reads, "Using the following relations between the elements of scattering phase matrix, $P_{ij}$(i,j=1,2,3,4), and the elements of amplitude matrix, $S_i$ (i=1,2,3,4), below,

$$|S_4(\theta)|^2 + |S_2(\theta)|^2 = (P_{11}(\theta) + P_{12}(\theta)) \times C_{sca}, \qquad (6)$$

$$|S_4(\theta)|^2 - |S_2(\theta)|^2 = (P_{21}(\theta) + P_{22}(\theta)) \times C_{sca}, \qquad (7)$$

where $C_{sca}$ is the scattering cross-section of a particle. As described above, the CASPOL detects light over single band of back scattering angles. 168° to 176°. To compare to the CASPOL measurements, we define the mean modeled depolarization ratio over the angular range of 168° to 176° and is expressed

5   below in Eq. (8).

$$\bar{\delta}_{Model}(168°:176°) = \frac{\int_{168°}^{176°}(P_{11}(\theta) + P_{12}(\theta) - P_{21}(\theta) - P_{22}(\theta)) \sin(\theta)\, d\theta}{2\int_{168°}^{176°}(P_{11}(\theta) + P_{12}(\theta)) \sin(\theta)\, d\theta} \qquad (8)"$$

*Referee Comment:*

It would also potentially be helpful for the authors to further discuss the use of the T matrix model for

10   dust (and ice)? A recent technical note (Koepke et al., ACP, 2015, 5947) may be helpful. Generally, the paper would be enhanced with some additional details, clarity or references (and/or possibly even information in the experimental section) associated with the model calculations.

*Authors' response:* To clarify, the ice crystal calculations were performed using improved geometric optics methods, while the dust calculations were performed using t-matrix. We have now added additional

15   details regarding each of these methods, and have added more additional references.

*Authors' changes in the manuscript:* The text on pg. 14 ln 4-13 now reads, "To compute the scattering phase matrices of these models with specific sizes at CASPOL wavelength, we apply so-called improved geometric optics method (IGOM) for particle with relatively large size and the invariant imbedding T-matrix method (II-TM) for particles with relatively small sizes (Yang et al., 1996; Bi et al., 2013; Bi and

20   Yang, 2014; Johnson, 1988). The combination of these two methods is chosen because of the different size parameters of the aerosol and ice crystal populations. The T-matrix method is a highly accurate

method for calculating scattering properties of atmospheric particles (Koepke et al., 2015; Brooks et al., 2004). However, it becomes impractical for large particles due to its excessive demands on the computational power. In contrast, the IGOM is accurate over the range of particle sizes over which the particle size to be much larger than the incident wavelength (Xu et al, 2017)."

Added references:

Brooks, S. D., O. B. Toon, M. A. Tolbert, D. Baumgardner, B. W. Gandrud, E. V. Browell, H. Flentje, and J. C. Wilson (2004), Polar stratospheric clouds during SOLVE/THESEO: Comparison of lidar observations with in situ measurements, *Journal of Geophysical Research-Atmospheres*,

10    *109*(D2).10.1029/2003jd003463.

Koepke, P., J. Gasteiger, and M. Hess (2015), Technical Note: Optical properties of desert aerosol with non-spherical mineral particles: data incorporated to OPAC, *Atmospheric Chemistry and Physics*, *15*(10), 5947-5956.10.5194/acp-15-5947-2015.

Xu, G., B. Sun, S. D. Brooks, P. Yang, G. W. Kattawar, and X. Zhang (2017), Modeling the inherent optical properties of aquatic particles using an irregular hexahedral ensemble, *Journal of Quantitative Spectroscopy & Radiative Transfer*, *191*, 30-39.10.1016/j.jqsrt.2017.01.020.

20    *Referee Comment:*
Overall, there is a lack of consistency within the text and figures where attention to detail would help. This is true, especially with the ordering of the types of particles within the different sections and also within the figures and captions. Further axis labels should include units where possible. A specific example is that in Fig. 6, there are both model and experimental results displayed but the y-axis includes

25    the model label and the x-axis is missing units. Some additional specifics are included below.
*Authors' response:* Thank you for addressing these specific inconsistencies. We have edited many of the figures in response to this comment and others.

*Authors' changes in the manuscript:* Please see the manuscript for revised figures. Labels and captions are now be consistent.

Specific comments:

5 *Referee Comment:* Pg 9, line 24: e is missing from the

*Authors' response:* Added.

*Referee Comment:* Pg 10, line 11: "both" is unnecessary and confusing.

*Authors' response:* Deleted.

*Referee Comment:* Pg 10, line 19-20: In final copy, watch for placement of the minus sign

Pg 11, line 6-7: placement of training

*Authors' response:* This has now been fixed.

*Referee Comment:* Pg 11, line 8: based on the figure, the authors mean total backscatter vs. depolarization ratio. I'd also suggest reversing the order in the follow up sentence on lines 8-10.

*Authors' response:* The text in this section (now pg 12, ln 14-17) has been revised has been revised for clarity.

*Referee Comment:* Pg 12, line 7: k is in eqn 3, but omega and t are not present. Is the equation missing time dependence? Also, r is not defined until line 11. Pg 12, equation 4: related to above, are both matrices and matrix elements included? Pg 12, missing comma after Pij

*Authors' response:*

25 Yes, indeed Equation, 3 should be written as

$$\boldsymbol{E}_i = \begin{pmatrix} E_{\|i} \\ E_{\perp i} \end{pmatrix} e^{i(kz-\omega t)} = \begin{pmatrix} E_{\|i} \\ 0 \end{pmatrix} e^{i(kz-\omega t)},$$

but note that $e^{i(\omega t)}$ is later omitted when express the relation between incident and scattered field in Eq(4) , because the scattering is assumed to be elastic. As in the original, r is introduced when it first used. *Authors' changes in manuscript.* Equation 3 has been corrected and "…scattering is assumed to be elastic." is now included. The elements of the amplitude matrix in Equation 4 are now defined: $S_i$

5 (i=1,2,3,4) in Eq(4). Also, the text as been added: "Note that $e^{i(\omega t)}$ term is omitted since the scattering is assumed to be elastic."

*Referee Comment:* Pg 12, missing comma after Pij
*Authors' response:* Added. Thank you.

*Referee Comment:* Pg 14, line 5: I think you mean Fig. 3a here.
*Authors' response:* Corrected. Thank you.

*Referee Comment:* Pg 14, lines 14-17: This is confusing, please clarify. In figure 3b, it seems the % of

15 total population of all particles having a depolarization ratio of 0.2 is close to 100%. How do other ratios exist for the population? This also makes interpretation of values in the text confusing.
*Authors' response:* Please note that the log y-scale is percent, not fraction. At a depolarization of 0.2 the percent of particles is ~1 %, not 100 %. Note that the log y-scale is percent, not fraction.
Authors' changes in the manuscript: None.

*Referee Comment:* Fig 6: Does it make sense to include the error bar information in the caption to make the figure less busy? Or at least remove and caption some of it?
*Authors' response:* We agree that the figure is busy, but the error bars reported are a big part of our discussion so it's important to retain these in the figure.

25 Authors' changes in the manuscript: No change has been made.

*Referee Comment:* Pg 17, line 1: typo of added "u". Also here you switch from >< notation to larger than and smaller than.

*Authors' response:* The text has been modified as suggested. "Larger than" and "smaller than" are now used throughout the text.

*Referee Comment:* Pg 18, line 12: typo likely

5 *Authors' response:* Corrected.

*Referee Comment:* Pg 20, line 4: double check wording for how this figure is introduced and also in the caption to be consistent and correct

*Authors' response:* The figure has been modified and now labels M as the "multiplication factor" as stated

10 in the text and caption.

*Referee Comment:* Pg 20, line 5: suggest figure or Fig. 7

*Authors' response:* Corrected to "Figure"

15 *Referee Comment:* Fig 8: Caption could be improved, especially repetition in description of panel c.

*Authors' response:* Agreed and corrected.

*Authors' changes in manuscript:* The caption now reads "Figure 8: Application of depolarization ratio method on three CFDC runs. Aerosol composition and temperature are labeled in the title. (a) Time series of supersaturation with respect to water. (b) INP Concentrations under normal (blue) and WDBT (red)

20 conditions are shown for the traditional (circles) and new (asterisks) analysis methods. (c) The normalized number distributions of all particles detected by the CASPOL. Time is reported in local time (CET)."

*Referee Comment:* Pg 20, line 26, suggest: In 2 out of 3 cases shown. Alternatively, you may want to clearly state (as you do later) that 27 cases/periods were evaluated (see Fig 9).

25 *Authors' response:* This is a good suggestion. Done.

*Referee Comment:* Pg 21, line 7: center panel of Fig. 8c?

*Authors' response:* Changed to "middle" panel.

*Referee Comment:* Pg 21, line 12: is data missing from fig 9 or is it just hard to see?

*Authors' response:* We have confirmed that no data is missing here. There are cases where no WDBT conditions occur in a run so there is no data to report.

5  *Authors' changes in the manuscript:* On page 22 ln 5, this is now noted in the manuscript, "In cases 24, 25, and 26 WDBT did not occur, so no data is reported.

*Referee Comment:* Fig 10: Which axis contains the data for the new method? I suspect the x, but am unsure due to confusion noted above. Please clarify and update axes. Would it make sense to fit this data

10  to observe is there is a small bias in the new data?

*Authors' response:* The figure has now been correctly labeled with the traditional concentration on the x-axis and the new concentration on the y-axis. Though a fit could be used to describe the bias, the authors felt that a new discussion about a completely different application of a linear regression would be confusing to readers. The object of the plot is achieved by discussing the biases of the new method.

*Referee Comment:* Pg 22, paragraph beginning on line 5: I am confused about how the errors in two regions can be 500 and 50% but overall it's 32%. I believe this is averaged values for each region considered. Is this the best way to present the uncertainty? Also as a minor detail, spacing when reporting numbers is inconsistent here and somewhat throughout the document, which would probably be fixed

20  upon typesetting.

*Authors' response:* Fit to the results of the linear regression (Eq. 9), which has a very large y-intercept contribute to this variable performance. The 500% represents the lowest range of concentration detected and represents just a small portion of the large range of INP concentrations measured during the FIN-02 campaign. Above the 50,000 $L^{-1}$, the new method's performance improves greatly with measurements are

25  within 50 % of the traditional concentration. No measurements here have error larger than 50%. The mean error for all measurements is 32.1 %.

*Authors' changes in the manuscript:* We've reordered and modified the text slightly to make this clearer.

*Referee Comment:* Figure 11: Consistency with previous figures and also double check captioning.

*Authors' response:* Thank you. The caption is now consistent with other figures.

*Referee Comment:* Pg 23, line 9: Does Fig 11b warrant more of a discussion? Can a literature comparison
5  be included?

*Authors' response:* This case was discussed earlier in the text, and does deserve an additional brief discussion here. Since a detailed FIN-02 intercomparison is forthcoming in the literature in the near future (DeMott, et al, 2017), we limit the discussion here.

[revised manuscript text omitted]